# Genome-wide microhomologies enable precise template-free editing of biologically relevant deletion mutations

Janin Grajcarek [1], Jean Monlong [2,5], Yoko Nishinaka-Arai [1], Michiko Nakamura[1], Miki Nagai[1], Shiori Matsuo[1], David Lougheed [3], Hidetoshi Sakurai[1], Megumu K. Saito [1], Guillaume Bourque [2,3] & Knut Woltjen [1,4]*

The functional effect of a gene edit by designer nucleases depends on the DNA repair outcome at the targeted locus. While non-homologous end joining (NHEJ) repair results in various mutations, microhomology-mediated end joining (MMEJ) creates precise deletions based on the alignment of flanking microhomologies (µHs). Recently, the sequence context surrounding nuclease-induced double strand breaks (DSBs) has been shown to predict repair outcomes, for which µH plays an important role. Here, we survey naturally occurring human deletion variants and identify that 11 million or 57% are flanked by µHs, covering 88% of protein-coding genes. These biologically relevant mutations are candidates for precise creation in a template-free manner by MMEJ repair. Using CRISPR-Cas9 in human induced pluripotent stem cells (hiPSCs), we efficiently create pathogenic deletion mutations for demonstrable disease models with both gain- and loss-of-function phenotypes. We anticipate this dataset and gene editing strategy to enable functional genetic studies and drug screening.

[1] Center for iPS Cell Research and Application (CiRA), Kyoto University, Kyoto 606-8507, Japan. [2] Department of Human Genetics, McGill University, Montréal, QC H3A 0G4, Canada. [3] Canadian Center for Computational Genomics, Montréal, QC H3A 0G1, Canada. [4] Hakubi Center for Advanced Research, Kyoto University, Kyoto 606-8501, Japan. [5] Present address: UC Santa Cruz Genomics Institute, University of California, Santa Cruz, CA 95064, USA. *email: woltjen@cira.kyoto-u.ac.jp

Editing genomic DNA with designer nucleases can have various functional outcomes depending on the endogenous repair pathway recruited and type of mutation created at the targeted locus (Fig. 1a). Classical non-homologous end joining (c-NHEJ), which directly re-ligates DNA ends following a double strand break (DSB), has been attributed to the efficient formation of various insertions and deletions (indels), and has thus been most commonly employed for gene disruption[1–3]. Yet, varying indels can result in disparate phenotypic outcomes ranging from silent mutations to the gain-of-function or loss-of-function (GOF or LOF) of a protein[4]. In contrast, DSB repair (DSBR) by homology directed repair (HDR) leads to outcomes dictated by homologous templates. HDR may be used for precise gene editing, albeit at low efficiencies without selection[5]. An alternative pathway, microhomology-mediated end joining (MMEJ), involves end-resection and the alignment of microhomologous sequences flanking DSBs. Subsequent removal of the intermediate sequence results in deletions of predictable size and identity[6]. Recently, MMEJ has been shown to play an important role in DSB repair, even in cells competent for c-NHEJ and HDR pathways[6,7]. The ability to create precise DSBR outcomes at a target locus would increase the efficiency and applicability of gene editing.

DSBR patterns can be influenced by a variety of factors including cell type[8–11] and cell cycle[12]. However, sequence context surrounding the DSB is the best understood determinant. DSBs induced by designer nucleases like transcription activator-like effector nucleases (TALENs) and CRISPR-Cas9 show replicable non-random repair patterns often based on the existence of microhomologies (μHs) neighboring the DSB[8–11,13,14]. Bioinformatic tools have been developed that predict the prevalent repair outcomes at a target site based on the presence of μHs and their distance from the DSB (heterology), and design of CRISPR-Cas9 guide RNAs (gRNAs) to introduce DSBs between μHs has been suggested as a strategy to knockout genes[10,15] or to correct mutations[14]. Still, the artificial introduction of a precise frameshift mutation alone does not guarantee a functionally predictable or even biologically relevant gene edit.

Deletions are the second largest class of mutations associated with disease, accounting for 25% of human pathogenic genetic variants annotated in the ClinVar database[16]. The observation that certain locus-specific pathogenic copy number variations (CNVs) and oncogenic chromosomal rearrangements are flanked by μH implicates MMEJ in contributing to deletion and translocation variants in humans[6,17]. Nevertheless, a systematic analysis of naturally occurring μH-flanked genetic variants in the human genome has yet to be completed.

In our current research, we create a tool called MHcut to uncover the extent to which μH-flanked deletion mutations can be found across the human genome. The μH-flanked variants identified by MHcut in 57% of all human deletion mutations, are candidates for recreation by harnessing precise and template-free MMEJ. We apply this gene mutagenesis strategy to create isogenic disease models in human induced pluripotent stem cells (hiPSCs), which are uniquely suited for this purpose, as they are able to differentiate into any tissue cell type[18,19] and are predisposed to MMEJ during DSBR[11]. Beyond demonstrating the value of MMEJ precision in functional genomics, we provide a resource of genome-wide μH-flanked deletion variants for further functional genetic studies.

## Results

### The majority of human deletion mutations are flanked by μHs.
To first determine the extent to which μH is associated with deletion variants in the human genome, we devised a tool called MHcut which searches for homologous sequences at the ends of annotated deletion variants (Fig. 1b,c, left). We applied MHcut to genomic location and sequence data of deletion variants annotated in dbSNP and ClinVar, the latter database containing valuable information on the clinical significance of mutations. MHcut aligns the 5′ and 3′ edges of each possible breakpoint that would result in the annotated deletion to the current reference human genome (GRCh38) and checks for maximum matching sequence (Fig. 1b, left). The minimum requirement for MMEJ has been proposed to be as few as 2 bp[9,15]. However, since analysis of CRISPR-Cas9 deletion data suggests that no clear correlation exists between repair outcomes following a DNA DSB and homologous sequences less than 3 bp[10], we opted for a more stringent minimum length of 3 bp of μH to classify deletion variants ≥3 bp in length as being μH-flanked. Despite observations that imperfect μH can lead to deletions of identical length yet differing sequence matching either the 5′ or 3′ residual μH[13], we only considered consecutive sequence matches as μH in order to conform specifically to annotated variant sequences.

Nearly 44 million deletion variants are represented in the current versions of the dbSNP and ClinVar databases (Fig. 1c, left). Strikingly, of the 19.3 million deletion mutations with a minimum size of 3 bp, 57% (11.1 million) are flanked by perfect μHs of at least 3 bp size, far exceeding the probability expected by random base distribution ($0.25^3 = 1.56\%$). Surprisingly, for deletions of at least 1 or 2 bp size with at least 1 or 2 bp flanking μHs, homologous bases are detected in 75% (expected $0.25^1 = 25\%$) and 67% (expected $0.25^2 = 6.25\%$) of variants, respectively (Supplementary Fig. 1a), implicating microhomology as a common enriched characteristic of human annotated deletion variants. We found a bias in μH-flanked variants towards short deletions of 3 or 4 bp (Fig. 1d) with similarly sized μH (Fig. 1e), such that 62% of variants have a heterologous sequence distance of zero between them (Fig. 1f) revealing that μHs are usually directly abutted. Regarding imperfect μH, allowance for a 1 bp mismatch following a perfect stretch of 3 bp or more and proceeding at least another single match only extended the homology length for 16% of variants (Supplementary Fig. 1b). The variant number could be only slightly extended to 11.9 million, or 62% of all deletions, if the definition of μH was relaxed to allow at least one stretch of continuous μH of 3 bp or more, emphasizing that 78% of variants are flanked by perfect μH (Supplementary Fig. 1c, d). The 11.1 million μH-flanked deletions ≥3 bp were found to be evenly distributed across all chromosomes (Supplementary Fig. 1e), with the vast majority residing in intronic or intergenic locations (Supplementary Fig. 1f). Still, 98,848 variants can be found in exonic regions, with 88% of all protein coding genes containing at least one μH-flanked deletion mutation in an exon (Fig. 1g), making large functional genomic screens feasible. The fact that less than 0.1% of μH-flanked deletions are annotated for clinical significance shows that there is an outstanding need for further functional exploration (Supplementary Fig. 1g).

### MHcut selects CRISPR sites for genome editing.
In the interest of generating more natural and disease-relevant mutations with high efficiency, we further developed MHcut to select candidates for creation by CRISPR-Cas9 cleavage and subsequent repair by MMEJ. To orchestrate MMEJ deletions, MHcut searches the local sequence of selected μH-flanked deletion variants for CRISPR-Cas9 protospacer adjacent motifs (PAMs) positioned so as to cause a DSB between the flanking μH, while retaining at least 3 bp of μH on either side of the break (Fig. 1b, c, right). MHcut then evaluates the upstream protospacer sequences of appropriately positioned PAMs to act as unique gRNAs in the reference human

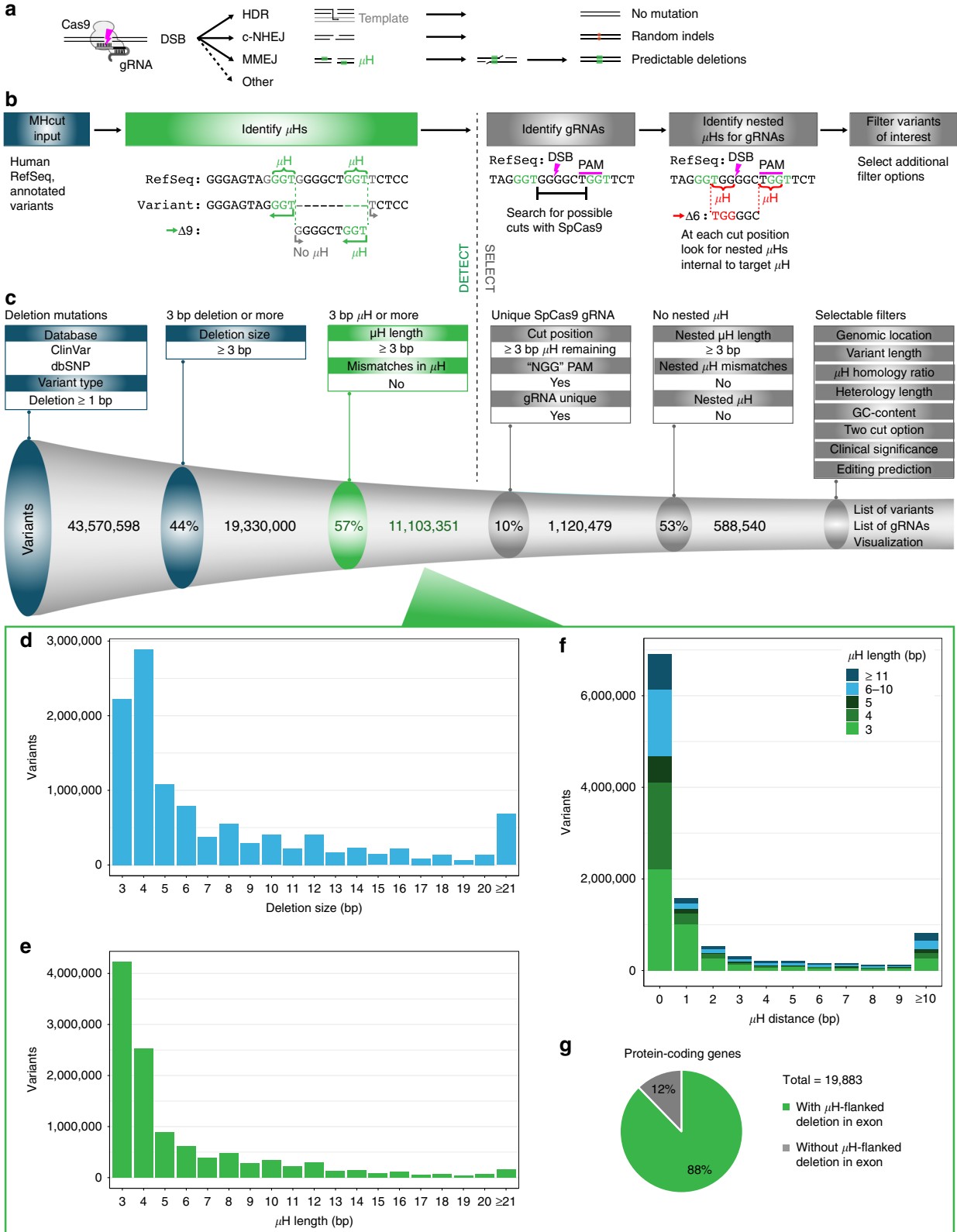

**Fig. 1** Analysis of human deletion alleles for flanking microhomologies (μHs). **a** Overview of double-strand break (DSB) DNA repair pathways and their outcomes. homology-directed repair (HDR), classical non-homologous end joining (c-NHEJ), microhomology-mediated end joining (MMEJ). **b** Schematic of the MHcut tool used to identify the microhomologous sequences. microhomologies (μHs, green) or nested μHs (red), SpCas9 PAM (underline) and DSB location (pink bolt). **c** Numbers for variant data input and result of filters applied to the MHcut tool output. bp, base pairs. **d** Distribution of μH-flanked deletion variants by deletion size. **e** Distribution of μH-flanked deletion variants by μH length. **f** μH-flanked deletion variants plotted by μH distance with μH length indicated by fill color. **g** Percentage of protein-coding genes with μH-flanked deletion variant in exonic coding sequences

genome. We initially chose to consider only 'NGG' sequences, the canonical *Streptococcus pyogenes* Cas9 (SpCas9) PAM, since SpCas9 represents the most commonly used and adaptable nuclease with a well-characterized cleavage site +3 bp upstream of the PAM[20]. Amongst the 11.1 million variants, 10% could be targeted with a unique SpCas9 gRNA (Fig. 1c, right), matching the predicted probability of 'GG' in positions +/−5, 6 on one side of the deletion (12.5%) for abutted μH, yet biasing the data set towards variants with more distant μHs due to the higher probability of identifying internal 'NGG' sites and unique gRNAs. Of the 10% of variants (1,120,479) that could be targeted with a unique SpCas9 gRNA, 3% are in exons (33,986). Of these variants, 33% or 11,168 deletions would result in a frameshift. Of note, 95% of these are variants of unknown significance (VUS). PAM requirements may be modified in MHcut in order to accommodate engineered SpCas9 variants (or alternative CRISPR/Cas systems introducing a blunt-ended cut) and expand the number of targetable variants. For example, allowing for engineered xCas9 with a relaxed PAM requirement targeting 'NG', 'GAA' and 'GAT'[21], increases the targetable number of variants to 33%.

For each gRNA and DSB site identified, MHcut also checks for μHs concealed inside of the annotated deletion variant (Fig. 1b, right). This step allows for the voluntary exclusion of variants with nested μHs that could theoretically reduce the efficiency of the desired deletion pattern, as μH with shorter intervening heterology are expected to be used preferentially[10,13,22]. An initial test at a locus in the GLA gene associated with Fabry disease revealed that nested μHs indeed reduce the efficiency of the targeted repair pattern (Supplementary Fig. 2a, b). Removing all variants with nested μHs further reduces the candidate list to about half (Fig. 1c, right). Additional filters are available to select variants of interest and associated gRNAs based for example on genomic location, clinical significance and prevalence of target editing outcome as predicted by the inDelphi tool[14]. The output of the tool with all filter options can be accessed online at https://mhcut-browser.genap.ca/ (Supplementary Fig. 3a, b).

**The creation of μH-flanked deletion variants is efficient.** To test if the loci identified by MHcut can indeed be repaired by MMEJ to reproduce the patterns found in humans, we chose a small set of candidate variants for proof-of-concept. The filter criteria for targets included the availability of a NGG PAM and unique gRNA for SpCas9, as well as pathogenic clinical significance, with a view to creating demonstrable disease models. From the short-list of 363 identified candidate variants (Fig. 2a), we chose targets with short μH distances, as is representative of the overall dataset, with varying μH lengths (Fig. 2b). Targets located on the X-chromosome were selected to simplify genotyping of CRISPR mutations in male ES and iPS cell lines.

Initially, a plasmid vector expressing SpCas9 and a single gRNA was transfected into HEK293T cells or 1383D6 hiPSCs (Supplementary Fig. 4a). Genomic DNA from the cell population was isolated, and the target locus PCR amplified and sequenced. The TIDE tool[23] was used to deconvolute the pool of sequences, revealing DNA repair patterns (Supplementary Fig. 4b). While the overall indel ratio for HEK293T cells was almost double that of hiPSCs at all targets (Supplementary Fig. 4c), likely reflecting transfection efficiency, the ratio of the target MMEJ mutation was on average one third higher in hiPSCs compared to HEK293T cells (Supplementary Fig. 4d). Apart from the KDM6A gene variant, the target MMEJ ratio in hiPSCs ranged between 70–100%. These results confirm that hiPSCs are especially suited for a gene mutagenesis strategy relying on MMEJ repair compared to a conventional cancer cell line.

With the goal of efficiently and accurately creating disease models, we optimized our CRISPR-Cas9 transfection protocol in human pluripotent stem cells. For that purpose, 1383D6 hiPSCs and H1 human embryonic stem cells (hESCs) were transfected in 96-well format with ribonucleoprotein (RNP) complexes containing recombinant SpCas9 and one gRNA specific to each target (Fig. 2c). Using RNP has the advantage of an acute activity and reduced off-target cleavage compared to plasmid transfection[24]. After 48 h, genomic DNA from the population was extracted and analyzed as described above. While the overall indel ratio was lower in H1 hESCs compared to 1383D6 hiPSCs, the indel ratios consistently showed that overall mutagenicity varied between the different targets (Fig. 2d). Importantly, the target MMEJ ratio was highly similar between hESCs and hiPSCs where, apart from the variant in the KDM6A gene, MMEJ ratios ranged between 70–100% (Fig. 2e). The low MMEJ ratio in KDM6A may be influenced by a low GC-content (33%), 'CA' dinucleotide repeats, or a T in position 4 upstream from the PAM, favoring +1 bp insertions[9,11]. Although indel ratios at each target differed between plasmid and RNP delivery, the target MMEJ ratios were comparably high in pluripotent stem cells (Supplementary Fig. 4c, d). In order to verify the functional relevance of these deletion variants, we cloned hiPSCs with target MMEJ mutations in DYSF, ALAS2 or FECH to establish disease models.

**DYSF knockout by MMEJ disrupts protein expression.** The DYSFERLIN transmembrane protein encoded by DYSF affects muscle fiber repair through regulation of membrane fusion events[25,26]. Various DYSF allelic variants are pathogenic, resulting in the LOF conditions Miyoshi muscular dystrophy 1 and limb-girdle muscular dystrophy type 2B, where patients experience progressive muscle weakness and atrophy. After concluding that our DYSF target faithfully repaired by MMEJ in the pattern of the natural variant (rs398123777), we established three DYSF-5bpDel mutant clones made with the plasmid-based approach (Supplementary Fig. 4a). All three selected clones are homozygous for the 5 bp deletion (Fig. 3a, Supplementary Fig. 5a). The DYSF-5bpDel mutation is predicted to be a LOF allele, as it causes a frame shift (Gln957Profs) and a stop codon at amino acid position 967 resulting in premature protein truncation (Fig. 3b). The clonal DYSF-5bpDel cell lines were confirmed to have retained their pluripotency as shown by OCT3/4 and NANOG immunostaining (Supplementary Fig. 5b) and to have no large chromosomal abnormalities in comparison to parental 1383D6 hiPSCs by SNP array (Supplementary Fig. 5c).

In order to observe the phenotypic consequences of the DYSF-5bpDel MMEJ deletion, we introduced a *piggyBac* transposon carrying the dox-inducible MYOD gene and mCherry reporter into hiPSCs, selected neomycin resistant populations, and differentiated them into a myogenic cell population over 5 days of dox treatment (Fig. 3c)[27,28]. A previously established Miyoshi myopathy patient hiPSC line served as a negative control[27] and parental 1383D6 hiPSCs served as a positive isogenic control. Between 50–60% of each *piggyBac*-transposed hiPSC population was mCherry positive indicating MYOD expression (Fig. 3d, Supplementary Fig. 5d). On day 7, populations were immunostained for Myosin Heavy Chain (MHC) to verify myogenic differentiation (Fig. 3d). Whereas the parental isogenic hiPSCs showed strong DYSFERLIN expression, all DYSF-5bpDel clones showed a complete loss of DYSFERLIN similar to the patient hiPSC line.

**A MMEJ deletion in ALAS2 causes gain-of-function phenotype.** Deletion variants are not always expected to result in gene knockouts, but rather have the potential to generate phenotypes

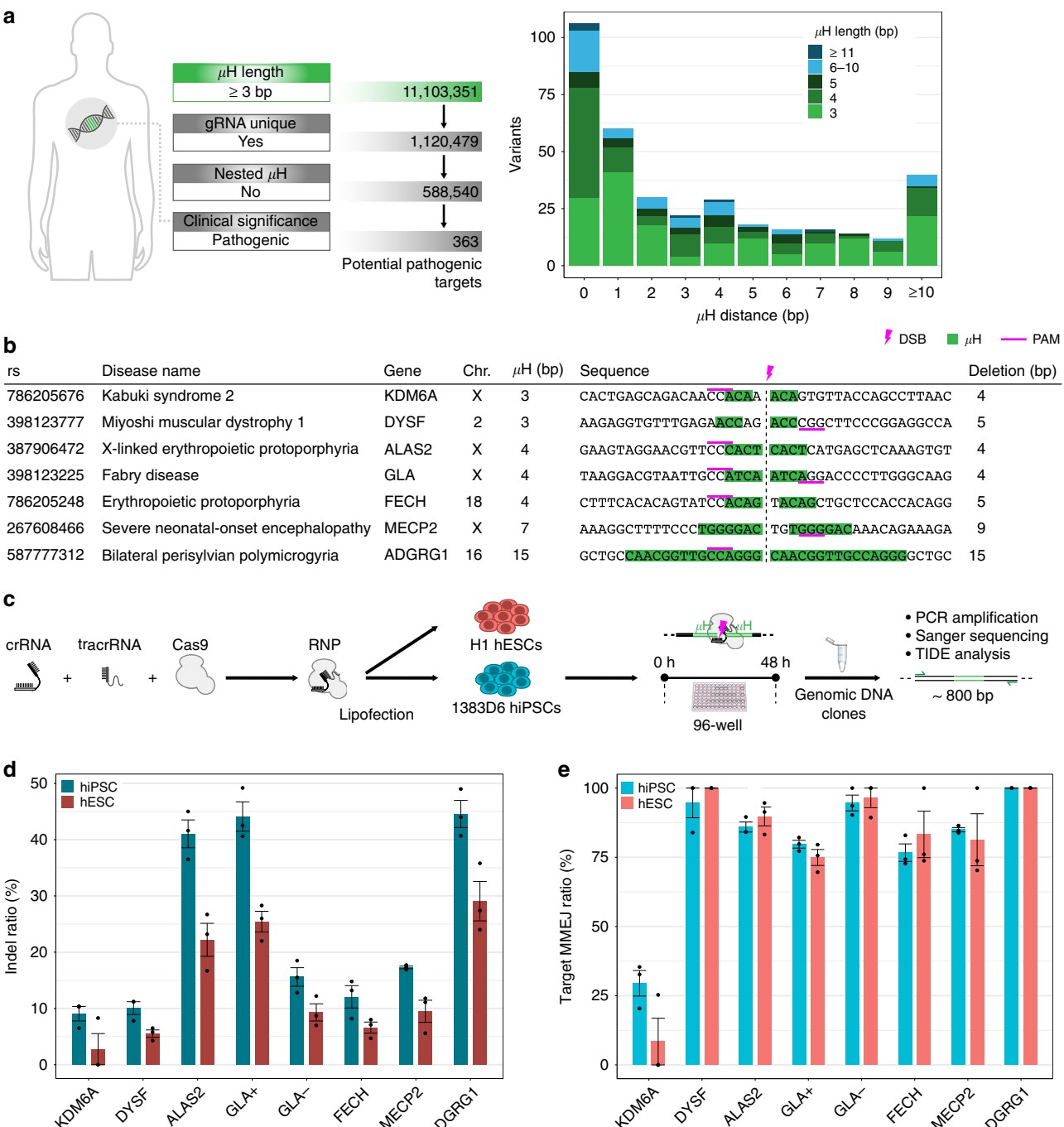

**Fig. 2** Selected pathogenic target μH-flanked deletion mutations can be recreated with high precision in hiPSCs and hESCs. **a** Filtered MHcut tool output of potential target pathogenic variants for the parameters shown. Graph at the right shows the distribution of target variants by μH distance with μH length indicated by fill color. **b** Selected target variant list. μH (green), DSB location (pink bolt), SpCas9 PAM (underline). **c** Schematic of the experimental method used to create MMEJ deletion alleles in 1383D6 hiPSCs and H1 hESCs. **d** Overall ratio of indel mutations found in the transfected hiPSC or hESC cell populations. **e** Ratio of the target MMEJ outcome among total indels. Means ± s.e.m. for $n = 3$ biological replicates. Source data are in the Source Data file

across a broad spectrum. In order to demonstrate this diversity in genotype/phenotype correlation, we established disease models for two target genes encoding enzymes in the heme synthesis pathway. In erythroid cells, 5′-Aminolevulinate Synthase 2 (ALAS2) is the first enzyme of the heme synthesis pathway producing 5-Aminolevulinic acid (5-ALA) (Fig. 4a). In other cell types, the paralog ALAS1 is expressed exclusively. 5-ALA is processed to Protoporphyrin IX (PPIX) by further enzymes. Ferrochelatase (FECH), the last and rate limiting enzyme of the pathway, integrates iron into PPIX to form heme. The ALAS2-

4bpDel mutation (rs387906472) causes a frame shift (Glu569-Glyfs) and extension of the ALAS2 protein C-terminus (Fig. 4b) resulting in a GOF allele with increased ALAS2 enzymatic activity leading to an overproduction of 5-ALA and PPIX[29]. FECH cannot process the resulting excess PPIX, which leads to its accumulation (Fig. 4a). Patients with the rs387906472 variant have abnormally elevated levels of PPIX in erythroid cells causing photosensitivity of the skin and in extreme cases liver damage, a condition referred to as X-Linked Protoporphyria (XLP)[29,30].

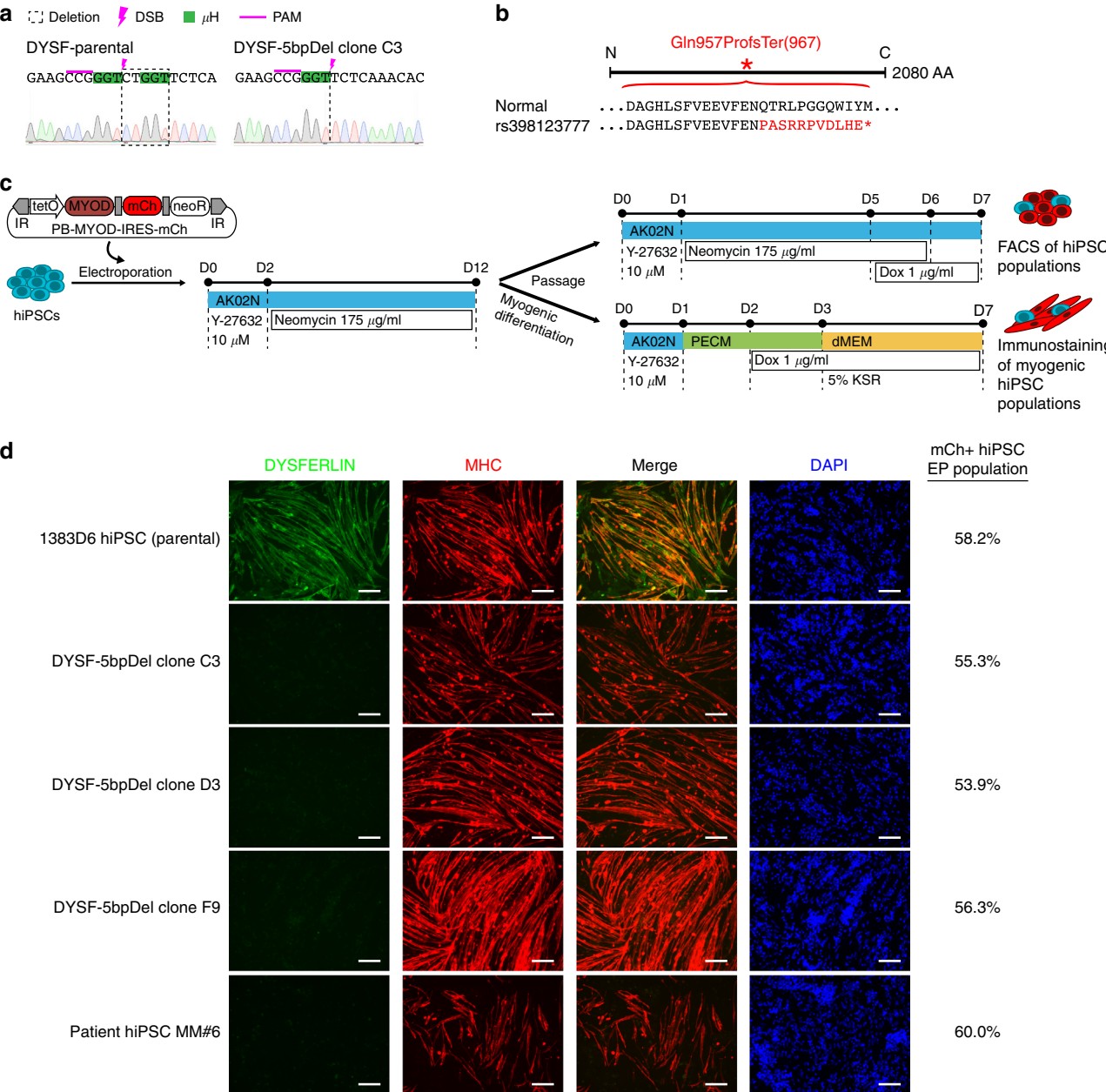

**Fig. 3** @Target variant associated with muscular dystrophy created by MMEJ recapitulates DYSFERLIN loss-of-function. **a** Sequence verification of a precise 5 bp deletion mutation in DYSF. Deletion (dotted line), DSB location (pink bolt), µH (green), SpCas9 PAM (underline). **b** Altered protein sequence of DYSFERLIN caused by the 5 bp deletion mutation. **c** Schematic of the experimental procedure to induce myogenic differentiation in hiPSCs by overexpression of MYOD from a *piggyBac* transposon. Differentiation day (D). **d** Immunostaining for DYSFERLIN and MHC in differentiated hiPSC populations. Comparison of the isogenic parental cell line, three derived clones carrying the disease mutation and a muscular dystrophy patient derived hiPSC cell line. Scale bar indicates 100 µm; ratio of mCherry + cells measured by FACS in corresponding electroporated (EP) hiPSC populations indicated on the right

@We selected six clones for the ALAS2-4bpDel mutation, three made with RNP and three made with plasmid transfection. All clones were confirmed to have the precise MMEJ mutation identical to the rs387906472 deletion variant (Fig. 4c, Supplementary Fig. 6a), to be OCT3/4 and NANOG positive by immunostaining (Supplementary Fig. 6b) and to have no large chromosomal abnormalities in comparison to parental 1383D6 hiPSCs by SNP array (Supplementary Fig. 6c). Since the ALAS2 gene encodes the erythroid-specific form of the 5-Aminolevulinate synthase[31,32], we differentiated ALAS2-4bpDel hiPSC clones to erythroid cells following an established protocol, with modifications (Fig. 4d)[33,34]. On day 26 of the differentiation, expression of the mature erythroid cell markers CD71 and CD235a was checked by FACS and activation of ALAS2 expression was confirmed by qPCR (Fig. 4e, f). The CD71 and CD235a double positive cell populations of the six ALAS2-4bpDel mutant clones, ranging from 70–90% of the total population (Fig. 4e), were shown by FACS to accumulate PPIX compared to the parental isogenic hiPSC (Fig. 4g). In addition, for 5 analyzed ALAS2-4bpDel clones, PPIX extraction from whole cell populations by ethyl acetate/acetic acid[35] showed a 14-fold to 30-fold increase of PPIX content compared to erythrocytes from isogenic parental hiPSCs (Supplementary Fig. 6d). In conclusion, the ALAS2-4bpDel mutant hiPSC clones faithfully recapitulate the expected XLP gain-of-function phenotype in vitro.

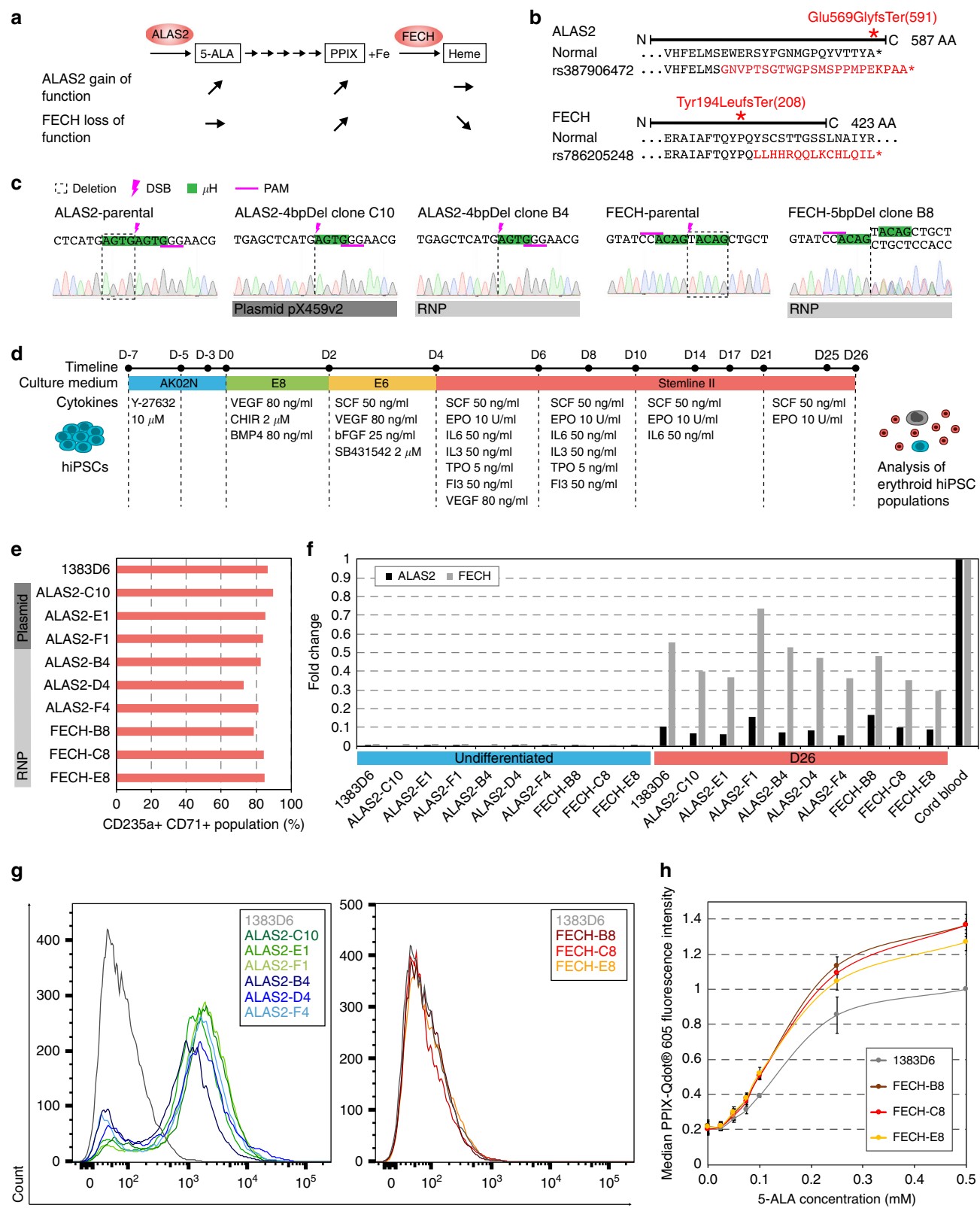

**A MMEJ deletion in FECH causes loss-of-function phenotype.** Finally, we established a LOF disease model for FECH, the last enzyme in the heme synthesis pathway (Fig. 4a). The rs786205248 deletion variant results in a frame shift (Tyr194-Leufs) and premature stop codon at amino acid position 208 rendering the protein non-functional (Fig. 4b)[36]. Patients with

the rs786205248 variant have a condition called Erythropoietic Protoporphyria (EPP). Similar to XLP, EPP is associated with photosensitivity of the skin and an increased risk of liver damage due to an accumulation of PPIX in cells (Fig. 4a). In humans this particular mutation of FECH is autosomal dominant, and can only be found in a heterozygous conformation. Indeed, in our

**Fig. 4** Erythropoietic protoporphyria disease models created by MMEJ display ALAS2 gain-of-function and FECH loss-of-function. **a** Phenotypic consequences of ALAS2 gain-of-function and FECH loss-of-function mutations on the accumulation of metabolites in the heme synthesis pathway. 5-Aminolevulinic acid (5-ALA), Protoporphyrin IX (PPIX). **b** Altered protein sequences of ALAS2 and FECH caused by deletion mutations. **c** Sequence verification of a precise 4 bp deletion mutation in ALAS2 and a 5 bp deletion in FECH generated by either plasmid or RNP transfection. Deletion (dotted line), DSB location (pink bolt), μH (green), SpCas9 PAM (underline). **d** Schematic for cell culture conditions during erythroid differentiation. Differentiation day (D). **e** Ratio of erythroid cells in differentiated hiPSC populations on D26, as indicated by CD235a-FITC and CD71-APC markers, measured by FACS. **f** Gene expression levels measured by qRT-PCR of ALAS2 and FECH in undifferentiated cells and D26 cell populations normalized to cord blood cells. **g** PPIX-Qdot® 605 fluorescence intensity in erythroid cell populations of mutant and normal cell lines on day 26, measured by FACS. **h** Dose-response curve of PPIX accumulation in response to 5-ALA treatment in undifferentiated hiPSC disease model clones and the isogenic parental line. Median fluorescence intensity of PPIX-Qdot® 605 measured by FACS and normalized to the parental cell line at 0.5 mM 5-ALA. Means ± s.d. for $n = 3$ biological replicates. Source data are in the Source Data file

experiment no hiPSC clones presented homozygosity for the 5 bp deletion mutation or the second most common mutation, a 1 bp insertion (Supplementary Table 1), either of which induce distinct frameshifts leading to premature stop codons. Our data suggest that FECH mutations are generally homozygous lethal in hiPSCs, which may account for the reduced indel ratio observed (Fig. 2d). Three clones made by RNP transfection were confirmed to be heterozygous with one normal allele and the precise MMEJ deletion on the other allele (Fig. 4c, Supplementary Fig. 6a). All three clones retain their pluripotency as shown by OCT3/4 and NANOG immunostaining (Supplementary Fig. 6b) and to have no large chromosomal abnormalities in comparison to parental 1383D6 hiPSCs by SNP array (Supplementary Fig. 6c).

Following differentiation of the FECH-5bpDel clones to erythrocytes (Fig. 4d, e) and observing an upregulation of ALAS2 and FECH expression (Fig. 4f), we could not detect increased levels of PPIX compared to erythrocytes from normal isogenic parental hiPSCs, neither by FACS (Fig. 4g) nor by plate reader (Supplementary Fig. 6d). This is consistent with the results found in EPP patient fibroblasts[37], and suggests that the FECH enzyme amount produced by heterozygous mutant cells is still sufficient to process PPIX under in vitro conditions. In addition, insufficient maturation of the heme synthesis pathway for in vitro differentiated erythroid cells may complicate comparison to in vivo disease conditions. Nevertheless, we could trigger a PPIX accumulation phenotype in undifferentiated FECH-5bpDel hiPSC clones by stressing the heme synthesis pathway through supplementation with additional 5-ALA in the culture medium[37]. The amount of PPIX in the cells increased in a dose-dependent manner as shown by FACS (Fig. 4h). In the absence of 5-ALA and at 0.025 mM 5-ALA, only background fluorescence was detectable. Starting from 0.05 mM 5-ALA, PPIX accumulation was detectable in all cells. A noticeable difference between the normal isogenic parental hiPSCs and the FECH-5bpDel clones was detected from 0.1–0.5 mM 5-ALA, where all three FECH-5bpDel clones had higher accumulation of PPIX ranging from 22–37% (Fig. 4h). This result, taken together with the fact that no homozygous hiPSC could be obtained, indicate that our MMEJ deletion alleles recapitulate the EPP phenotype.

**Assessing the scope of targetable variants**. The length of heterology is known to negatively affect the efficiency of MMEJ repair, with suggested limits ranging from 5 bp[10] to around 15 bp[11]. While over 80% of the variants in the MHcut dataset are either directly abutted or have a very short distance of 0–2 bp (Fig. 1f), the strict need for a unique Cas9 gRNA biases the set of targetable variants towards larger distances, with only 45% of variants having a μH distance of 0–2 bp. We expect this bias to be mitigated by using engineered Cas9 or orthologues with different PAM requirements.

To assess the impact of increasing μH-distance on the efficiency of template-free creation of the MHcut variants, we first analyzed the MHcut dataset with the inDelphi editing prediction tool[14]. The mean prevalence predicted by inDelphi (average score from available mESC, HCT116, HEK293, K562, U2OS cell data) for all MHcut variants with available Cas9 gRNA indicates that, depending on μH length, μH distances between 4 and 10 bp are expected to cause the MMEJ ratio of the target deletion to drop below 20% on average (Supplementary Fig. 7a).

To validate this data in hiPSCs, we chose a set of candidate variants with μH distances between 3 and 10 bp (Supplementary Fig. 7b) from our short-listed pathogenic variants (Fig. 2a), and transfected RNP complexes as described above (Fig. 2c). While we could not identify a clear limit for μH-distance impacting the target MMEJ ratio, we observed the highest MMEJ ratio (91%) for a novel variant in TSC2 with a μH distance of 7 bp (Supplementary Fig. 7d). As observed for GLA (Supplementary Fig. 2a), these data suggest that larger μH distances are more likely to contain nested μHs, interfering with target deletion creation. Importantly, the results also show that low gRNA efficiencies are another major obstacle in creating the target deletion (Supplementary Fig. 7c, d), supporting the need for reliable gRNA prediction software or engineered Cas9 variants to fully explore the MHcut dataset.

## Discussion

Although recent research into nuclease mediated DSBR outcomes has highlighted the importance of the MMEJ pathway[6,7], the extent to which μH can be found at naturally occurring deletion junctions had not yet been systematically investigated. Considering the importance of sequence context in predicting mutation outcomes[8–10,14,15], knowledge of extant μH can greatly enhance biological relevance, functionality, efficiency and precision of gene edits. While DNA repair outcome prediction tools, like inDelphi[14], use sequencing data resulting from artificially introduced DSBs to train their machine learning algorithms, our MHcut tool takes a different approach of analyzing the sequence of existing deletion variants in the genome for the presence of μH. In this sense, our research complements the available prediction tools (Fig. 5), by enabling researchers to reference mutations detected in the human population to modify their target gene, instead of introducing an artificial gene edit. Meanwhile, the prediction algorithms allow for the selection of the MHcut candidate variants with the highest probability for efficient recreation. The MHcut dataset can also contribute a relevant list of endogenous target loci for creating training datasets for prediction algorithms in the future. Here, we not only uncover the extent to which deletion mutations in the human genome are flanked by μH, which to our knowledge was not previously possible with currently accessible tools, we also prove the feasibility of our gene editing approach using DSBR by MMEJ to recreate the annotated disease alleles in hESCs and hiPSCs. Using this method, new disease models including appropriate isogenic

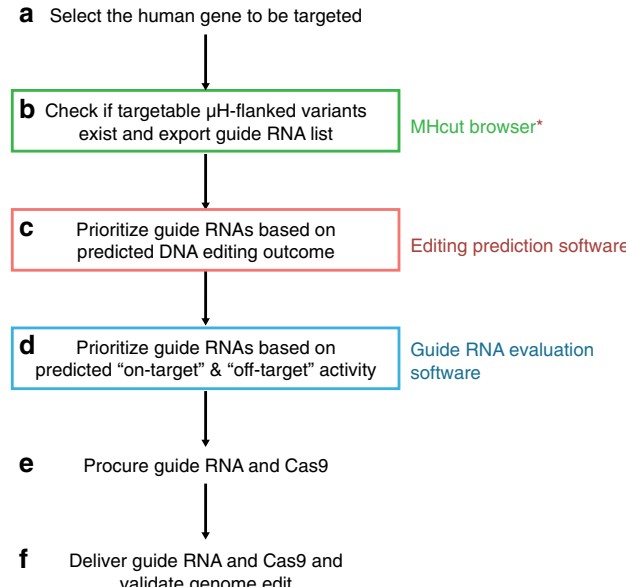

**a** Select the human gene to be targeted

**b** Check if targetable µH-flanked variants exist and export guide RNA list — *MHcut browser\**

**c** Prioritize guide RNAs based on predicted DNA editing outcome — *Editing prediction software*

**d** Prioritize guide RNAs based on predicted "on-target" & "off-target" activity — *Guide RNA evaluation software*

**e** Procure guide RNA and Cas9

**f** Deliver guide RNA and Cas9 and validate genome edit

**Fig. 5** MHcut tool complements existing tools for selecting a suitable gene editing target and gRNA. (*MHcut tool contains the editing prediction for the target deletion from the inDelphi editing outcome prediction tool[14])

controls can be established efficiently and quickly with gRNAs suggested by the MHcut tool.

MHcut identified 11 million deletion mutations of 3 bp or more that are flanked by µH of at least 3 bp or more. However, only 10% of variants are expected to be targetable by SpCas9. The limited number of gRNAs per MHcut variant and the prevalence of the target editing outcome, especially for variants with longer µH distances, are a constraint on the scope of targetable variants. We found gRNA efficiencies to be highly variable and therefore recommend prioritizing the gRNAs identified by the MHcut tool not only with tools for editing outcome prediction, but also with state-of-the-art gRNA on-target and off-target efficiency evaluation software (Fig. 5) to maximize the efficiency of target variant creation. Using other engineered CRISPR-Cas9 variants with flexible PAM sites[38,39], as well as TALENs or Zinc-finger nucleases (ZNFs), the genome-wide target space could be significantly increased. In addition, the overall number of candidate variants with µH of at least 3 bp or more could be further increased from 11 million by relaxing the criterion for mismatches in the µH. In order to accommodate disease mutations flanked by µHs with mismatches[6], the MHcut tool can be adjusted to allow for a customizable number of mismatches.

The fact that the majority of deletion mutations existing in the human genome are flanked by µHs hints at the extent to which MMEJ repair might contribute to genetic variation in humans. It has been demonstrated that MMEJ operates in the context of c-NHEJ and HDR competent cells. Other research groups have found gRNAs that induce a specific MMEJ deletion in up to 80–90% of mutant alleles in vertebrate cells[10] and up to 30–40% in HEK293T cells[9,10]. Our research in pluripotent stem cells shows that up to 100% of SpCas9-induced mutations can be attributed to a specific µH, but appear dependent on sequence features such as µH length and distance. Although in vitro results cannot retrospectively prove the natural cause of these variations, the phenomenon observed by our MHcut analysis suggests that MMEJ might be more prevalent in human cells than initially expected[6,7,40]. Based on this data trend, we suspect that additional µH-flanked variants might be found in sequencing data amassed

by efforts such as the 100,000 Genomes Project, Genome in a Bottle, SwissVar or the GenomeDenmark platform.

Hereafter, disease models created in hiPSCs by MMEJ can be used for drug screening and functional analysis in an isogenically-controlled genetic background. Concretely, the effect of Succinylacetone on the accumulation of Protoporphyrin[41] could be tested in our XLP erythroid cells. Moreover, high MMEJ ratios with up to 100% accuracy found for longer µHs suggest that applications for gene therapy are possible, especially since MMEJ has been shown to work under in vivo conditions in vertebrates[10]. Potential targets for gene therapy by MMEJ would result in alleles overcoming disease phenotypes caused by microduplications like familial hypercholesterolemia[14], frameshift mutations like Duchenne muscular dystrophy[42], or dominant mutant alleles like in XLP.

In the future, the uncovered variety of naturally occurring mutations flanked by µH will act as a resource to help expand our understanding of parameters influencing DSBR pathway choice (e.g., µH length, deletion length, µH distance, genomic location, etc.). In addition, they can be utilized to elucidate the function of DNA repair pathway components. Inhibiting or enhancing individual components can accelerate research into MMEJ pathway genetics and kinetics, opening up the possibility of further increasing the efficiency of MMEJ induced mutations[9,14,43] for rational application.

## Methods

**MHcut tool**. For a deletion entry, the variant's sequence and both flanking sequences are retrieved. Microhomology (µH) is tested for both flanking configurations: 5′ flanking region with 3′ variant sequence and 3′ flanking region with 5′ variant sequence. µH is extended as much as possible allowing for at most one mismatch. Both the full µH and the first stretch of exact µH is saved. Summary metrics on the µH include homology, distance between µHs and GC content. The µH is scored as the sum of matches in the full µH and the length of the first stretch of exact µH. The flanking configuration with the strongest score is then selected for further analysis. Of note, a variant of the tool was implemented to report results for both flanking configurations but showed minimal differences. In many cases the same deletion can be represented by different coordinates and can be shifted without affecting the deletion represented. For this reason, MHcut will also shift each deletion as much as possible and pick the coordinates that result in the longest µH.

Once the µH is defined, we select potential gRNAs. First, we look for PAMs that would result in a cut within the variant sequence, excluding the first 3 bp of the µH sequence. To filter cuts that would lead to large heterologies in large deletions, only cuts within 50 bp of the µHs are considered. Of note, this output still allows to design experiments with two cuts, one close to each µH, to recreate large deletions. For each position found, we save the PAM, strand, protospacer sequence and distance between the cut and both µHs.

Second, each gRNA is then tested for exact off-target candidates (0 mismatches) in the genome. The protospacer sequence and PAM are aligned to the genome using Jellyfish[44] for a fast exact alignment. In the absence of off-target candidates, the guide and PAM sequences should align to a unique position in the genome.

Finally, MHcut also checks for µHs nested inside of the annotated deletion variant, that could be preferred over the target µH, using the Bae et al. script[15] for each possible cut position. An exact µH on each side of a cut, at least 3 bp long and closer to each other than the target µHs is considered to be a nested µH. The number of nested µHs and strength are recorded for each guide and summarized at the variant level.

In addition, MHcut calculates the predicted prevalence of each variant among repair outcomes using the inDelphi algorithm[14] for each cut position and for each of the 5 cell types provided by inDelphi. The predicted prevalence is recorded for each guide and summarized for the best guide at the variant level.

MHcut outputs contain results at the variant level with µH metrics and the number of guides found at the different stages of the workflow, and at the guide level with information about guides with no perfect off-targets (Supplementary Table 2).

**dbSNP and ClinVar analysis**. Deletions were extracted from dbSNP[45] (October 28, 2018 release) and ClinVar[46] (July 1, 2018 release) on the GRCh38 genome version. The databases were downloaded from ftp://ftp.ncbi.nih.gov/snp/organisms/human_9606_b151_GRCh38p7/VCF/All_20180418.vcf.gz and ftp://ftp.ncbi.nih.gov/pub/clinvar/vcf_GRCh38/clinvar_20181028.vcf.gz.

After merging both databases, the variants were annotated according to their overlap with the gene annotation from Gencode v28[47], as either exonic, intronic, within UTRs, or intergenic. The dataset consisted of 43.7 M deletions once duplicates had been removed. We then used the MHcut tool described above to identify potential microhomology and CRISPR cuts that could recreate the deletions. Due to the scale of the input data, we used Jellyfish[44] (v2.2.6) to test for off-target guides. The main results presented here were produced for SpCas9's PAM (NGG) and allowing at most one consecutive mismatch in the microhomology. Eventually, MHcut was also run on the full dataset when using xCas9's PAMs (NGN, GAA or GAT)[21]. We created the pathogenic category in clinical significance by summarizing all variants that contain the word pathogenic in their label. Data analysis was conducted in R and Excel.

**MHcut browser data portal**. The variant-level and guide-level output were parsed and inserted into a PostgreSQL database. The portal interface was written in JavaScript with the D3 library, using a Python server written with the Flask framework to query the database. Thorough indexing and caching of results ensured that the queries scaled to millions of data points. The data portal is available at https://mhcut-browser.genap.ca/.

**Cell culture**. HEK293T cells (Thermo Scientific, USA) were maintained in culture medium containing DMEM, 10% FBS, penicillin-streptomycin, and L-glutamine. HEK293T cells were passaged every 3–4 days.

Undifferentiated hESCs and hiPSCs were cultured under feeder-free conditions[48]. H1 hESCs[49] (WA01, WiCell, USA) and 1383D6 hiPSCs[50] (Center for iPS Cell Research and Application, Japan) were maintained in StemFit AK02N (AJINOMOTO) in 6-well or 96-well tissue culture dishes coated with Laminin iMatrix-511silk (Nippi), or iMatrix-511 (Nippi), respectively (0.5 µg/cm²). During the passage, addition of 300 µL of Accumax (Innovative Cell Technologies) for 6-well or 30 µL for 96-well and incubation at 37 °C for 15 min, followed by dissociation through pipetting were used to detach the cells. Cells from each surface were collected in 700 µL or 100 µL medium containing 10 µM ROCK inhibitor, Y-27632 (Wako). The Countess II FL Automated Cell Counter (Thermo Fisher Scientific) was used to count cells using trypan blue exclusion. For a typical cell passage, $1 \times 10^4$ cells were seeded in a 6-well plate in media containing Y-27632. ROCK inhibitor was removed on the second day of culture. On the seventh day after passage, cells reached 80% confluency and were passaged again. Frozen stocks were prepared by resuspending cells at $1 \times 10^6$ viable cells per 1 mL STEM-CELLBANKER (Takara) and freezing 200 µL cell suspension in a cryogenic tube in liquid nitrogen. One vial of cell stock was defrosted per well of a 6-well plate in media containing Y-27632. Cells were regularly tested for mycoplasma.

To induce Protoporphyrin IX (PPIX) accumulation in hiPSCs, undifferentiated 1383D6 cells, and derived FECH-5bpDel clones, 5-Aminolevulinic Acid (5-ALA) was added to the cell culture medium at 0.025 mM, 0.05 mM, 0.075 mM, 0.1 mM, 0.25 mM, and 0.5 mM concentrations. Medium was changed daily until the cells were analyzed using flow cytometry on the fifth day as described below.

**Plasmid construction**. Supplementary Table 3 provides a list of sequence-verified plasmids used in this study. For plasmid gene editing, sgRNA oligonucleotides were annealed and cloned into pSpCas9(BB)-2A-Puro V2.0 plasmid (pX459v2, a gift from Feng Zhang, Addgene plasmid #62988) linearized with BbsI as described by Ran et al.[51]. In brief, equal volumes of 100 µM oligos were annealed by heating to 95 °C for 5 min, followed by a slow cool down and a 1:200 dilution in water. 100 ng linearized vector were then mixed with 2 µL annealed oligo and 4 µL Ligation Mix (Takara) and incubated at 16 °C for 30 min. The mix was then transformed into competent bacteria using heat shock at 42 °C. Plasmid DNA from clones was extracted using the Wizard® Plus SV Minipreps (Promega) Kit according to the manufacturer's instructions. Primers used for cloning are listed in Supplementary Table 4. Sequences were verified using the U6 primer 'GAGGGCCTATTTCC-CATGATTCC' (dna790) and Sanger sequencing described below. Complete sequences are available upon request.

**Cell transfection**. For plasmid transfection into HEK293T cells, 1 µg of the pX459v2 vector containing the respective sgRNA were prepared in 150 µL OPTI-MEM I reduced-serum medium (Invitrogen) containing 6 µL Fugene HD transfection reagent (Promega). After 15 min incubation, 300 µL cell suspension containing $3 \times 10^5$ cells were added and cells were transferred to a 6-well culture plate. Genomic DNA was extracted three days after transfection.

For plasmid transfection into 1383D6 hiPSCs, 3 µg of plasmid were transfected by NEPA21 (Nepa Gene Co., Ltd) electroporation with 125 V and a poring pulse length of 5 ms into $1 \times 10^6$ cells in single-cell suspension in Opti-MEM medium (Thermo Fisher Scientific), as described[50]. Half of the electroporated cells were plated in a 60 mm cell culture dish well in medium containing Y-27632. On the day after electroporation, Y-27632 was removed and in the case of pX459v2 plasmids 0.5 µg/mL puromycin (Sigma-Aldrich) was added for 48 h. Genomic DNA was extracted nine days after transfection.

For RNP transfection H1 hESCs and 1383D6 hiPSCs were seeded 5 days before the transfection as described above. On the fifth day cells were harvested at 50–60% confluency and diluted to $2 \times 10^5$ cells per mL for reverse transfection using

Lipofectamin™ Stem transfection reagent (invitrogen) as per the manufacturers recommended protocol. Alt-R CRISPR-Cas9 V3 protein, synthetic tracrRNA, and crRNAs (Supplementary Table 5) were purchased from IDT. First, equimolar amounts of crRNA and tracrRNA in the provided duplex buffer were mixed and heated to 95 °C for 5 min and allowed to cool to room temperature to form the gRNA. Both gRNA and Cas9 protein were diluted to 1 µM in OPTI-MEM (Gibco). Subsequently, equimolar amounts of 500 ng Cas9 protein and gRNA were mixed in a final volume of 25 µL OPTI-MEM per reaction and incubated for 20 min at room temperature (RT) to form the RNP complex. The transfection reagent was diluted at 1 µL in 24 µl OPTI-MEM per reaction. RNP complexes were added to diluted transfection reagent and incubated for 20 min at RT. Per reaction in 96-well format, 50 µL transfection mixture were mixed with 100 µL cell suspension (20,000 cells) in media containing Y-27632. As a negative control cells were transfected with gRNA only or Cas9 protein only. Cells were cultured for 48 h before being harvested for analysis as described above.

For picking clones, part of each RNP-treated cell population was passaged into a 6-cm dish at clonal density (600 cells). Colonies were isolated manually with a micropipette and cultured in 96-well format as described above.

**Genomic DNA extraction**. Cells were harvested as described above. For genomic DNA extraction the cells were subsequently washed twice in PBS, before the DNA was extracted using different protocols.

Genomic DNA from plasmid transfections and selected hiPSC clones was extracted from $0.5–1 \times 10^6$ harvested cells using the DNeasy Blood and Tissue Kit (Qiagen) according to the manufacturer's instructions.

For sequence screening of hiPSC clones picked in 96-well format, genomic DNA was isolated by adding 50 µL plate lysis buffer[52] (10 mM Tris-HCl, pH 7.5, 10 mM EDTA, 0.5% Sarcosyl, 10 mM NaCl, and 1 mg/mL Proteinase K) and incubating overnight at 55 °C. Subsequently, 100 µL ice cold solution of 100% EtOH and 5 M NaCl were added for DNA precipitation. The samples were vortexed, incubated for about 1 h and then centrifuging at 9,100 xg for 15 min. After removing the supernatant and rinsing twice with 70% EtOH the DNA was re-suspended in TE pH 8.0, according to expected DNA concentration[52].

Cells transfected with RNP were cultured for 48 h and then lysed using EpiBio Quick Extraction solution (Epicenter). Briefly, cells were harvested as described above and transferred to a 96-well PCR plate. After washing twice with PBS, 50 µL of DNA extraction solution was added per well and the plate was vortexed vigorously for 15 s. After incubating the plate for 6 min at 65 °C, it was vortexed again, followed by a final incubation at 98 °C for 2 min. The lysates were used directly for PCR or stored at −30 °C.

**Sanger sequencing**. To verify the DNA repair outcome in the transfected cell populations and picked clones, a genomic region of 700–800 bp surrounding the target site was PCR amplified using the primers listed in Supplementary Table 6. Primers were designed using NCBI Primer-BLAST with optional settings for human repeat filter, SNP handling, and primer pair specificity checking to the H. sapiens reference genome. PCR was performed using Phusion Hot Start II DNA Polymerase (Thermo Scientific) on a Veriti 96-well Thermal Cycler (Applied Biosystems) according to the manufacturer's instructions. Specific PCR conditions are available upon request.

Prior to sequencing, PCR products were treated with ExoSAP-IT Express Reagent (Affymetrix) according to the manufacturer's instructions. The BigDye Terminator v3.1 Cycle Sequencing Kit (Applied Biosystems) was applied for DNA sequencing, followed by ethanol precipitation. Subsequently, the sequencing was performed on a 3130 × 1 Genetic Analyzer (Applied Biosystems). Sequence alignments were conducted using Snapgene v.3.1.4 or higher (GSL Biotech LLC). Sequence trace files with poor base calling confidence were excluded from further analyses.

**TIDE analysis**. To deconvolute the mixed sequences resulting from various DNA repair outcomes in the cell populations we performed TIDE analysis using the R script provided by the authors[23]. The deletion size window was extended to the maximum of 50 bp to accommodate larger deletions. The right boundary of the deconvolution window was adjusted individually by target to exclude low-quality sequence trace regions at the end of the amplicon. The p-value, calculated with a two-tailed t-test of the variance–covariance matrix of the standard errors[23], was adjusted to $p < 0.05$ from the default $p < 0.001$ to include possible rare repair events in the analysis. The remaining parameters were set to default or allowed to adjust automatically.

**SNP array**. Genomic DNA from iPSC clones modified by gene editing were genotyped using an Infinium OmniExpress-24 v1.2 and v1.3 (Illumina) SNP array according to the manufacturer's recommendations. Median probe spacing is around 2 kb. Array scanning was performed on an iScan Bead Array Scanner (Illumina). Scanned data were processed using Illumina GenomeStudio (2011.1 for v1.2 and 2.0.4 for v1.3) with human genome Build 37 and FinalReports were exported according to the manufacturer's protocol. CNV call was done using a combination of PennCNV[53] (1.0.3), GWASTools[54] (v1.2.R), and MAD[55] (1.0.1). Karyograms were prepared using GenomeJack (Mitsubishi Space Software).

**Myogenic differentiation**. Doxycycline-inducible MyoD-hiPSCs were generated and differentiated to myogenic cells as previously described[27,28] (Fig. 3c). In brief, hiPSCs were electroporated with 1 µg each of PB-TAC-ERN-hMyoD and pCAG-PBase, as described above. Two days after electroporation 175 µg/mL Neomycin was added to the medium. On day 12, cells were passaged.

To check the percentage of the population expressing MyoD, undifferentiated hiPSCs were induced for 48 h with 1 µg/mL doxycycline, before mCherry expression was confirmed by flow cytometry, as described below.

For myogenic differentiation, $3 \times 10^5$ of the dissociated MyoD-hiPS cells were seeded onto a Matrigel (BD Biosciences) coated 6 well-plate in AK02N medium containing Y-27632 at day 0. On day 1, culture medium was replaced by Primate ES Cell Media (Reprocell). On day 2, MyoD expression was induced by adding 1 µg/mL doxycycline. On day 3, the culture medium was changed to differentiation medium containing alpha Minimal Essential Medium (αMEM; Nacalai Tesque) with 5% KSR (Invitrogen) and 1 µg/mL doxycycline. Medium was changed daily until cells were used for immunohistochemistry on day 7.

**Immunohistochemistry of cultured cells**. For staining of the pluripotency markers NANOG and OCT3/4, medium sized hiPSC colonies were treated with a solution of 4% Paraformaldehyde (PFA) PBS for 10 min at room temperature (RT). Cells were permeabilized using a solution of 0.1% Triton X-100 in PBS for 10 min at RT, followed by a blocking step in 3% bovine serum albumin (BSA) in PBS for 60 min at RT. The primary antibodies, Nanog (D73G4) XP Rabbit mAb (#4903S, Cell Signaling) (1:500) and Purified Mouse Anti-Oct3/4 (#611203, BD Biosciences) (1:250) were prepared in blocking solution and cells were incubated overnight at 4 °C. After a washing step, cells were incubated for one hour at RT with the respective secondary antibodies Alexa Fluor 546 goat anti-rabbit immunoglobulin G (IgG) (H + L) (#A11010, Life Technologies) (1:500) and Alexa Fluor 546 goat anti-mouse IgG (H + L) (#A11030, Invitrogen) (1:500) prepared in blocking solution containing 1 µg/mL DAPI. After 3 subsequent washing steps, the cells were observed under a fluorescence microscope. Phase-contrast and fluorescence images were acquired on a BZ-X710 (KEYENCE) using appropriate filters and exposure times.

Differentiated myogenic cells were fixed with a solution of 2% PFA in PBS for 10 min at 4 °C, followed by a treatment with MeOH:H₂O₂ (100:1) at 4 °C for 15 min. The cells were blocked with Blocking One (Nacalai Tesque) at 4 °C for 1 h. Subsequently, samples were incubated with the primary antibodies Anti-Myosin Heavy Chain mouse monoclonal antibody (MF20, #14-6503, eBioscience) (1:800), Anti-Dysferlin rabbit monoclonal antibody (#ab124684, Abcam) (1:200) in 10% Blocking One/PBS solution containing 0.2% Triton X-100 at 4 °C overnight. After a washing step, cells were incubated for 1 h at RT with the secondary antibodies Alexa Fluor 568 Goat anti-Mouse IgG1 (#A21124, Thermo Fisher) (1:500) and Alexa Fluor 488 Goat anti-Rabbit IgG (#A11034, Thermo Fisher) (1:500) in 10% Blocking One/PBS solution containing 0.2% Triton X-100 and DAPI (1:5000). Samples were observed with a BZ-X700 (Keyence).

**Erythroid differentiation**. For differentiation towards erythroid cells the protcols by Ohta et al. and Niwa et al.[34,56] were modified as follows. Briefly, before the start of erythroid cell differentiation hiPSCs were prepared at clonal density (~400 cells) in a 10 cm cell culture dish and cultured for about 7 days as described above, until the colonies reached a size between 750–1000 µm in diameter. The erythroid differentiation (Fig. 4d) was started by changing the medium to Essential 8 (E8; Gibco) containing 80 ng/mL VEGFA (R&D system), 2 µM CHIR99021 (Merck) and 80 ng/mL BMP4 (R&D system). During the entire differentiation cells were kept at 37 °C in 5% O₂ hypoxic condition. After 2 days of culture, the medium was changed to Essential 6 (E6, Gibco) containing 50 ng/mL SCF (R&D system), 80 ng/mL VEGFA, 25 ng/mL bFGF (Wako) and 2 µM SB431542 (Milltenyi). From day 4 onwards, cells were cultured in Stemline II medium (Sigma) containing 50 ng/mL SCF and 10 U/mL Erythropoietin (EPO, Merck). In addition, from day 4 until day 21, 50 ng/mL IL-6 (R&D system) were added to the medium. Furthermore, from day 4 until day 10, samples were cultured in 50 ng/mL IL-3 (R&D system), 5 ng/mL TPO (R&D system) and 50 ng/mL Flt3/Flk2 (FI3, R&D system). Also, from day 4 until day 6 80 ng/mL VEGFA were added to the medium. On day 10 and day 17, cells were dissociated by pipetting through a cell strainer and passaged into a suspension culture at $5 \times 10^4$ cells/mL. On day 26, erythroid differentiation and Protoporphyrin IX (PPIX) expression were confirmed by flow cytometry and ALAS2 and FECH gene expression were measured by qRT-PCR as described below. In addition, the amount of PPIX was confirmed using ethyl-acetate/acetic acid extraction and resuspension in 1.5 mol/L HCl, as described[35]. The fluorescence of the cell extract was measured using the EnVision (PE) with photometric filters 590/7 (2100–5500; PE) and 420 nm optical filter (2100–5400; PE) and compared to a PPIX (Frontier Scientific) standard dilution curve.

**Flow cytometry**. For measurement of mCherry fluorescence intensities, $3.0 \times 10^5$ cells were resuspended in FACS buffer (PBS containing 2% FBS), filtered through a cell strainer and analyzed using a BD LSRFortessa Cell Analyzer (BD Biosciences) with BD FACS Diva software (BD Biosciences).

For the confirmation of erythroid differentiation, $1 \times 10^5$ cells were stained with anti-human CD235a-FITC (#349103, Biolegend, 1:20) and CD71-APC (#334107,

Biolegend, 1:20) in a total volume of 10 µL FACS buffer containing 0.5 µL Human TruStain FcX™ (biolegend). Isotype controls were conducted using FITC and APC Mouse IgG2a antibodies (#400209 and #400221, Biolegend). After a 20 min incubation on ice, the samples were washed with 100 µL PBS containing 0.5 µg/mL DAPI. Finally, the cells were resuspended in 100 µL FACS buffer and passed through a cell strainer for analysis on a BD FACS ARIA II Cell Sorter (BD Biosciences) with the BD FACS Diva Software version 8.0.1 (BD Biosciences).

For measurement of PPIX fluorescence intensities, which displays an autofluorescence with a peak at 632 nm when excited at 409 nm[57], the Qdot® 605 filter setting (excitation 405 nm, emission 610 nm + /− 20 nm) of the BD FACS ARIA II Cell Sorter was used.

Flow cytometry data were analyzed by FlowJo software v9.7.6 or higher (Tree Star).

**qRT-PCR**. Total RNA was isolated from bulk cell populations on day 26 of erythroid cell differentiation using the RNeasy Plus Mini Kit (Qiagen) according to the manufacturer's instructions. For cDNA synthesis, 100 ng of RNA were reverse transcribed using the SuperScript III First Strand Synthesis System (Invitrogen) following the manufacturer's instructions. For qRT-PCR analysis the SYBR Premix ExTaq II (Takara) was applied according to the manufacturer's instructions. The analysis was conducted using the QuantStudio 3 (Thermo Fisher Scientific). Human GAPDH was used to normalize the expression levels of the target transcripts. qRT-PCR primers are listed in Supplementary Table 7.

**Reporting summary**. Further information on research design is available in the Nature Research Reporting Summary linked to this article.

## Data availability

The results on the dbSNP and ClinVar deletions can be explored in the web application https://mhcut-browser.genap.ca/. The scripts from this analysis are available on the GitHub repository at https://github.com/WoltjenLab/MHcut and the dataset was deposited on Figshare at https://doi.org/10.6084/m9.figshare.9118364.v1. Source data is available in the Source Data file The data that support the findings of this study are also available from the corresponding author upon request. The pX459v2 plasmid can be accessed via Addgene (#62988).

## Code availability

MHcut is written in Python and available at the Python Package Index repository (PyPI), on Zenodo at https://doi.org/10.5281/zenodo.3403353 and on GitHub at https://github.com/WoltjenLab/MHcut. A Docker container is also provided.

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

## Acknowledgements

We would like to acknowledge the technical assistance of Ryoko Hirohata for plasmid construction, Tomoko Matsumoto for cell culture and Kanae Mitsunaga for FACS analysis. We extend our gratitude to Feng Zhang for provision of reagents through Addgene. Shiori Takehara performed preliminary experiments creating a μH-flanked variant in the GLA locus while supported by the CiRA Research Internship Program (2016). David Bujold supervised the development of the MHcut browser at the Canadian Center for Computational Genomics with support from Genome Canada. Data analyses were enabled by compute and storage resources provided by Compute Canada and Calcul Québec. The authors would like to thank Hiromi Dohi, Fumiyo Kitaoka, Masaki Nomura, Tomoko Takahashi, Masafumi Umekage, and Naoko Takasu for performing the SNP array analysis. This research was supported in part by grants to K.W. from the Cell Science Foundation (Japan), to G.B. from Fonds de Recherche Santé Québec (FRSQ-25348), to M.K.S. from AMED (17935400 and 179354230), and to H.S. from AMED (17bm0804005h0001). K.W. is a Hakubi Center Special Project Researcher.

## Author contributions

K.W. conceived and J.G. designed the study. J.G. performed the majority of experiments. J.G. and J.M. conceived the MHcut tool. J.M. programmed the MHcut tool with input from J.G., G.B., and K.W. J.G. and J.M. analyzed the output from the MHcut tool. M. Nakamura. supported sequencing and gene expression experiments. M.Nagai and H.S. performed myogenic differentiation and subsequent immunostaining, and H.S. analyzed the data. J.G., Y.N., and S.M. performed the erythroid differentiation under the super-vision of M.S. D.L. developed the MHcut browser. J.G. and K.W. wrote the paper with feedback from all authors. All authors approved the final paper.

## Competing interests

The authors declare no competing interests.
