## [Peer Review File · Nature Communications]

Reviewers' Comments:

Reviewer #1:

Remarks to the Author:

In this study, the authors describe a new approach for improving the accuracy and efficiency of CRISPR-Cas9 mediated gene knockout. They first note that the target sequences of many human deletion mutations are associated with flanking regions containing microhomologies (μ H), that μ H are found flanking pathogenic mutations, and in a significant minority of cases, are found within coding regions. They present a computer program that enables identification of these regions, and a general strategy to target CRISPR-Cas9 cleavage to the region between the μ H sites. In proof of concept studies in human iPSC, they show that precise deletions mimicking pathogenic mutations in several genes can be generated efficiently through μ H end-joining.

The concept of exploiting μ H in CRISPR mediated gene editing is not novel, but the concept of recreating naturally occurring mutants, combined with the computer program and the new strategy, expands the potential of this approach. The data are clearly presented and support the authors' conclusions. Their findings will be of general interest to those using iPSC for functional genomics and gene modeling, and as the authors note, the findings also have implications for our understanding of how deletion mutants arise. There are a number of specific questions relating to the extent of applicability of this technology that the authors should address.

Specific comments:

1. Page 5-How much allelic variation is observed at the μ H target sites identified using your program relative to the reference genome; that is how many of these μ H are likely to occur only in a fraction of the population? Maybe if the μ H is absent the individual would not be likely to suffer a deletion mutation.
2. Page 6-Did you attempt to estimate what proportion of the deletions within protein coding regions would be predicted to have consequences for protein function? This would be a larger category than those of known clinical significance, correct (Figure 2a)?
3. How many of the 10% of variants that could be targeted with unique gRNA were in exonic regions?
4. Page 7-Do the 363 candidates represent the totality of genes targetable with this approach? If so how much bigger would this number be with the various adjustments discussed?
5. Page 8-293 cells are aneuploid, would this have contributed to the comparison with diploid pluripotent stem cells?
6. Page 8-it is a bit of stretch for the authors to say that they have created a muscular dystrophy model. They have actually just confirmed that the deletion disrupts protein expression.

Reviewer #2:

Remarks to the Author:

This study by Woltjen lab reports the significant contribution of MMEJ in disease mutations containing deletions more than 2 bp, provides a tool for searching such MMEJ-dependent deletions, named MHcut, and shows some experimental examples to create pathogenic deletions in hPSCs. This is a follow-up study of recent inDelphi paper published in Nature, and likely becomes a good complementation, because inDelphi and MHcut focus on the detailed editing outcomes and genome-wide detection/annotation, respectively. In this context, this reviewer thinks the authors should try to connect these two studies more deeply, as described later.

Overall, the display items are clear and placed adequately, the manuscript is well written, and discussion and conclusion seem scientifically sound. However, regarding the experimental design, the population analysis should be more comprehensively conducted for further characterization and validation of the template-free MMEJ repair and the MHcut tool, although the clonal analysis

was sufficiently performed.

1) This reviewer imagines that researchers can search the target mutations using MHCut, and then predict the editing outcomes using inDelphi or other tools such as Microhomology-Predictor. If it is correct, this scheme should be presented in the manuscript for the practical research. In addition, although inDelphi paper is already cited a few times in the manuscript, the description is too brief and lacks important information such as the name "inDelphi" and the involvement of machine learning. Since the inDelphi study and this study are closely related, thorough introduction and discussion should be provided.

2) In addition to the above point, the activity and specificity of sgRNA are also quite important to choose the target site. Is this point considered in MHCut tool? As the authors may know very well, there are a number of tools developed for scoring the sgRNA activity and specificity. Is it possible to implement such a function in the MHCut tool?

3) In the population analysis displayed in Figure 2, the target sites containing various microhomology lengths resulting in various deletion lengths were examined. However, all of them contained less than three distances between two microhomologies. The current manuscript contains only one data for longer distance (Figure S2). It seems insufficient, because there are a wide variety of microhomologies distances (Figure 2a). Various population analysis should be added for the target sites containing longer distances. If it is difficult to efficiently create mutations with longer distance in a template-free manner, this point should be clearly described as a limitation of the authors' approach.

4) The actual usefulness and importance of some selectable filters are unclear. For example, two-cut option was provided, but there is no actual example in the manuscript. Regarding the GC-content, there is a suggestive description (line 225), but it is still unclear without showing other examples.

5) The statement "data not shown" is prohibited in Nature Communications (<https://www.nature.com/ncomms/submit/how-to-submit>). The authors are requested to provide all the data.

Reviewer #3:

Remarks to the Author:

In this work, the authors describe a comprehensive computational analysis of microhomologies in the human genome, where they are widespread (highlights: in exons in most protein-coding genes, and surrounding the majority of annotated deletion mutations). They describe a computational tool, MHCut, which is suggested as a method for identifying candidates for precise repair to a target deletion allele via microhomology-mediated end-joining (MMEJ). The authors demonstrate genotypic validation of predicted precision editing in 7 disease-relevant deletion targets, and characterize the efficiency of MMEJ in human induced pluripotent stem cells (hiPSC) compared to HEK293T cells. The authors also perform cursory functional assays on hiPSC cell lines generated from 3 disease-relevant precise deletion edits.

While some of their analysis is interesting, I think publishing this paper would mislead the community, as MHCut is a strictly inferior tool to several recently published algorithms aimed at predicting Cas9 outcomes (inDelphi, FORECasT from Shen et al., 2018 and Allen et al., 2018). The above-referenced algorithms have been trained and tested on extensive collections of Cas9 outcome profiles, while MHCut seemingly does not use training data and uses a modest set of testing data.

Based on the work described, we believe that the authors overstate in the abstract, introduction, and discussion the general extent to which MHCut is able to identify candidates, and the extent to which candidates exist in the human genome, for precise template-free editing. A representative

text is "Here we introduce a tool called MHCut that identified 11 million naturally occurring deletion mutations flanked by μ Hs across the human genome, covering 88% of protein-coding genes. These mutations are candidates for precise creation in a template-free manner by MMEJ repair." in the abstract — the implication is that a substantial fraction of these 11 million deletion mutations identified by MHCut would be precisely created experimentally, but the manuscript fails to support this claim.

If the authors believe that MHCut has advantages over inDelphi/FOREcst (which have user-friendly interfaces and can be easily utilized to give what are likely to be more accurate estimates of genomic sites capable of precise Cas9-induced repair to a mutation allele), then they should more clearly state and test such specific claims. Otherwise, it will only confuse the research community to have a published tool that has not been compared to the state-of-the-art, uses more vague terminology (the authors never define precise repair, although the above-referenced papers give clear, quantitative definitions) and is likely to be strictly inferior.

While I do not support revision of this manuscript for Nature Communications, below I list some technical suggestions on other aspects of the work:

1. One claim of particular note is that the majority of pathogenic deletions in humans possess microhomology. The authors should use simulated random deletion data to derive background rates of random deletion outcomes of particular lengths that would possess microhomology. This would allow them to calculate the statistical significance of their claim that microhomology deletions are enriched among human pathogenic deletions.
2. An analysis of what fraction of pathogenic deletions could be created at specific levels of precision (e.g. >50% of all genotypic outcomes) from inDelphi/FOREcst would be a helpful addition, as it would likely be much more accurate than MHCut and thus helpful to the community. This might also clarify the failure of precise deletion at the KDM6A locus, where the well-documented possibility of 1-bp insertions (Shen et al., 2018, Allen et al., 2018, Kalhor et al., 2018, Lemos et al., 2018, Taheri-Ghahfarokhi et al., 2018) confounds the precision of their targeted repair (the authors do not describe MHCut as considering the possibility of 1-bp insertions).
3. In the three experiments where the authors installed biologically relevant deletion mutations selected by their tool, the authors claimed that SNP arrays enabled them to confirm the absence of additional CNVs or large deletions in edited cells compared to the parental cell line, but the data is not shown. The authors' claims would be better supported if the data were shown. However, I have further concerns that a SNP array approach is insufficient to fully confirm the authors' claim. A SNP array queries only a tiny fraction of the genome, severely limiting the position and resolution of detectable deletion events. Furthermore, in a diploid setting, only deletions in heterozygous alleles would yield different SNP array results compared to the parental cell line. To better support the works' claim that useful disease models can be created by the authors' strategy, I would suggest additional assays for measuring large deletions and additional CNVs. In particular, large deletions at the expected cutsite could be detected by targeted long-read sequencing (PacBio, Nanopore) or long-range PCR (Kosicki et al., 2018).
4. The authors state that only a small fraction of deletion mutations have known phenotypes, yet all three mutations they model have known phenotypes. It would have been more interesting to investigate mutations without known functional effects. For example, the experiments in Fig. 3 are highly expected. Showing that a bi-allelic frameshift prevents protein expression is not a novel or interesting finding. To support their claim that precise, template-free deletions will enable novel approaches to disease modeling, the paper should include novel biological analysis and conclusions about at least one mutant allele created using this method.
5. Lastly, the authors note in the discussion that "Furthermore, the MHCut data and MMEJ could be applied to create protective alleles against communicable disease, including the well-characterized 32 bp deletion in CCR5 which confers resistance to HIV infection." This claim is poorly supported by the work described and is inconsistent with the literature on MMEJ. Among the 7 deletion targets described in the work, five are 4-5 nt deletions, and the longest is a 15-nt deletion. Though particular outcomes in MMEJ are favored if they have strong microhomology, a stronger factor is an exponentially decreasing frequency of repair of longer deletions. As a result, a 32-bp deletion is

expected to occur with much lower precision, and the authors fail to provide evidence against this expectation.

**NCOMMS-19-03896**

**Genome-wide microhomologies enable precise template-free editing of**
**biologically relevant deletion mutations**

**REVIEWERS' COMMENTS**

**Reviewer #1**

In this study, the authors describe a new approach for improving the accuracy and
efficiency of CRISPR-Cas9 mediated gene knockout. They first note that the target
sequences of many human deletion mutations are associated with flanking regions
containing microhomologies (μ H), that μ H are found flanking pathogenic mutations,
and in a significant minority of cases, are found within coding regions. They present a
computer program that enables identification of these regions, and a general strategy
to target CRISPR-Cas9 cleavage to the region between the μ H sites. In proof of
concept studies in human iPSC, they show that precise deletions mimicking
pathogenic mutations in several genes can be generated efficiently through μ H end-
joining.

The concept of exploiting μ H in CRISPR mediated gene editing is not novel,
but the concept of recreating naturally occurring mutants, combined with the
computer program and the new strategy, expands the potential of this approach. The
data are clearly presented and support the authors' conclusions. Their findings will be
of general interest to those using iPSC for functional genomics and gene modeling,
and as the authors note, the findings also have implications for our understanding of
how deletion mutants arise. There are a number of specific questions relating to the
extent of applicability of this technology that the authors should address.

We thank Reviewer #1 for recognizing the importance of the natural phenomenon we
identified, as well as the practical contribution it provides to the field of gene editing.

Please allow us to address each of the Reviewer's specific comments in turn below.

1. Page 5-How much allelic variation is observed at the μ H target sites
identified using your program relative to the reference genome; that is
how many of these μ H are likely to occur only in a fraction of the

population? Maybe if the μ H is absent the individual would not be likely to
suffer a deletion mutation.

We thank Reviewer #1 for raising this interesting question. Indeed, if a human iPS
cell line contains an allelic variation that would significantly alter the sequence of the
μ H, it would likely decrease the possibility of creating the targeted deletion.

To get an idea of how widespread this phenomenon could be, we tested a
random subset of 100,000 μ H-flanked variants identified by MHcut. We overlaid all
known common allelic variants (SNPs, deletions and insertions with MAF >1%,
except the variant of interest) downloaded from the UCSC Genome Browser
(<http://hgdownload.soe.ucsc.edu/goldenPath/hg38/database/snp141Common.txt.gz>),
with the genetic locations of the identified μ H-flanks. Within the sample data, 83% of
μ H-flanks do not overlap with any common allelic variation, whilst 8% overlap with
other deletions, 7% with SNPs, 1% with insertions and 2% contain multiple SNPs or
Indels (Rebuttal Fig. 1). Deletions or insertions are more likely to impact the μ H-
sequence significantly and might reduce the efficiency of creating the target deletion,
whereas SNPs that only cause a single mismatch within the μ H might be tolerated
(Kim S.I. et al., 2018, Ref. #13).

Overall, we believe the probability of the cell line of interest or the specific
patient to have an allelic variation in exactly the μ H-flanked variant to be created to
be low. Nevertheless, the tractability of any gene editing strategy is expected to be
subject to human genetic variation (Scott and Zhang, 2017, PMID: 28759051).
Therefore, it is common practice to compare the sequence of individual genomic loci
to the RefSeq before conducting editing experiments.

**Rebuttal Fig. 1:** Proportion of variants of a random subset of 100,000 μ H-flanked
 deletion mutations, whose μ H regions overlap a common (MAF >1%) SNP, an
 insertion/deletion, or a combination of those.

2. Page 6-Did you attempt to estimate what proportion of the deletions
 within protein coding regions would be predicted to have consequences
 for protein function? This would be a larger category than those of known
 clinical significance, correct (Figure 2a)?

Reviewer #1 raises an important point regarding the absence of functional information
 currently available for most gene variants. Of the 11 million variants with at least a 3
 13 bp μ H, only 5322 have an annotated clinical significance (Supplementary Fig. 1g), of
 14 which 2040 are in the pathogenic category (Fig. 2a).

In order to estimate the number of deletions within protein coding regions
 with possible consequences for protein function, we analyzed how many of the
 98,848 exonic variants (Supplementary Fig. 1f) have a variant length not divisible by
 three, which would likely cause a frameshift. The result is 26,323 variants, or 27% of
 all exonic variants. Note that this excludes 3528 variants overlapping splice
 acceptor/donor sites, and deletions, which do not cause frameshifts, but are expected
 to disrupt key protein domains. The type of effect the frameshift might have on
 protein function needs to be confirmed on a case by case basis.

As Reviewer #1 anticipated, the number of μ H-flanked deletion variants that
 could have clear consequences for protein function is more than 5 times larger than
 the number of variants with annotated clinical significance, supporting a need for

continued functional validation. To highlight this point, we added a graph to
**Supplementary Figure 1f**

**Supplementary Fig. 1f. Genomic location of μH -flanked deletion variants and**
**proportion of exonic variants resulting in a frameshift on the right; un-translated**
**region (UTR); variant length (varL).**

3. How many of the 10% of variants that could be targeted with unique
gRNA were in exonic regions?

We thank Reviewer #1 for raising another interesting question. We have added this
information to the Results section (page 6, lines 4-7) as follows: “Of the 10% of
variants (1,120,479) that could be targeted with a unique SpCas9 gRNA, 3% are in
exons (33,986). Of these variants, 33% or 11,168 deletions would result in a
frameshift. Of note, 95% of these are variants of unknown significance (VUS).”

4. Page 7-Do the 363 candidates represent the totality of genes targetable
with this approach? If so how much bigger would this number be with the
various adjustments discussed?

We thank Reviewer #1 for the opportunity to clarify the number of targetable variants.
The 363 candidates shown in Figure 2a represent only the variants with annotated
pathogenic clinical significance targetable with a unique Cas9 guide RNA and that
contain no nested μH to increase potential repair outcome efficiency. The overall
number of variants targetable with this approach, including variants of unknown
significance (VUS) is 588,540, as shown in Figure 1c. We recognize that the original

Figure 2a was not clear, and have revised the labelling to emphasize our selection of
pathogenic variants only for the proof-of-concept experiments. As Reviewer #1
correctly points out, the number of targetable variants could be increased, e.g. by
utilizing Cas proteins with relaxed PAM requirements. In the case of the xCas9
enzyme, the number of targetable variants with a unique guide RNA and no nested
μ H would be increased to 2,575,210. Of those variants, 1091 have a known
pathogenic clinical significance.

**5. Page 8-293 cells are aneuploid, would this have contributed to the**
**comparison with diploid pluripotent stem cells?**

We appreciate Reviewer #1's concern regarding the influence of cell ploidy on the
DNA repair outcomes. Ploidy should not impact the ratio of DNA repair pathway
outcomes to each other, as long as the alleles do not contain SNPs in the guide RNA
binding site or the microhomology sequence, which we confirmed with sequencing in
the cell lines used. The Cas9-RNP-complex introduces a DNA double-strand-break on
all available alleles, which are then repaired through endogenous repair pathways.
Therefore, an increase in the number of Chromosomes only increases the total number
of possible repair events, not the ratio of the contribution of e.g. MMEJ to the repair
outcomes. Should the RNP complex not be able to target all alleles, the number of
unedited alleles would increase, decreasing the overall Indel rate. However, the
MMEJ ratio of the mutant allele fraction would not be impacted.

**6. Page 8-it is a bit of stretch for the authors to say that they have created a**
**muscular dystrophy model. They have actually just confirmed that the**
**deletion disrupts protein expression.**

We agree with Reviewer #1. Indeed, we do not show that the loss of the DYSF
protein caused by the μ H-based deletion actually leads to the muscular dystrophy
phenotype. A causal relationship between the loss of DYSF protein and impaired
membrane repair observed in muscular dystrophy has been published by Tanaka A. et
al., in 2013 (Ref. #27), albeit for a different loss-of-function mutation. To help clarify
our currently presented data, we have modified the title of Figure 3 to "Target variant

associated with muscular dystrophy created by MMEJ recapitulates DYSFERLIN
loss-of-function." and changed the Result part sub-heading to "DYSF knockout by
MMEJ disrupts protein expression" (page 7, line 33). We hope that these changes are
satisfactory.

Reviewer #2

This study by Woltjen lab reports the significant contribution of MMEJ in disease
mutations containing deletions more than 2 bp, provides a tool for searching such
MMEJ-dependent deletions, named MHcut, and shows some experimental examples
to create pathogenic deletions in hPSCs. This is a follow-up study of recent inDelphi
paper published in Nature, and likely becomes a good complementation, because
inDelphi and MHcut focus on the detailed editing outcomes and genome-wide
detection/annotation, respectively. In this context, this reviewer thinks the authors
should try to connect these two studies more deeply, as described later.

Overall, the display items are clear and placed adequately, the manuscript is
well written, and discussion and conclusion seem scientifically sound. However,
regarding the experimental design, the population analysis should be more
comprehensively conducted for further characterization and validation of the
template-free MMEJ repair and the MHcut tool, although the clonal analysis was
sufficiently performed.

We thank Reviewer #2 for their fair evaluation of our manuscript, and raising the
importance of connecting our resource - a surveying existing mutations for
microhomology - and the inDelphi DNA repair prediction tool. We outline our
perspective and efforts to integrate inDelphi data in the points below.

1. This reviewer imagines that researchers can search the target mutations
using MHcut, and then predict the editing outcomes using inDelphi or
other tools such as Microhomology-Predictor. If it is correct, this scheme
should be presented in the manuscript for the practical research. In
addition, although inDelphi paper is already cited a few times in the
manuscript, the description is too brief and lacks important information

such as the name "inDelphi" and the involvement of machine learning.
Since the inDelphi study and this study are closely related, thorough
introduction and discussion should be provided.

We thank Reviewer #2 for the thoughtful suggestion of creating a scheme showing
the usage of MHcut tool in relation to other available prediction tools for practical
research. In order to clarify how the tools complement each other in this scheme, we
created Figure 5. In addition to the microhomology score (Bae et al., 2014, Ref. #15),
which was already part of the MHcut algorithm, we integrated the inDelphi prediction
(Shen et al., 2018, Ref. #14) as an average score from the 5 cell types tested for the
best guide RNA amongst those available for each variant in MHcut. All individual
scores for the target deletion for each guide RNA are listed in the corresponding guide
RNA file. This selectable filter was also added to Figure 1 C.

Moreover, we describe and mention the inDelphi tool by name in the Results
part (page 6, lines 19-22) and the Discussion part (page 12, lines 11-14) as follows:
"Additional filters are available to select variants of interest and associated gRNAs
based for example on genomic location, clinical significance and prevalence of target
editing outcome as predicted by the inDelphi tool¹⁴." and "In this sense, our research
complements existing tools (Fig. 5) that use machine learning to predict editing
outcomes at DSBs¹¹, like the inDelphi tool¹⁴, by enabling researchers to reference
mutations already existing in the human genome to modify their target gene instead of
introducing an artificial gene edit."

Unfortunately, human iPSC cells were not one of the cell types tested by
inDelphi, highlighting the need for a human iPSC-based training data sets in the
future. We believe that the MHcut resource can contribute a relevant list of
endogenous target loci for such an analysis.

Figure 5. MHcut tool complements existing tools for selecting a suitable gene editing target and guide RNA.
 (* MHcut tool contains the editing prediction for the target deletion from the inDelphi tool¹⁴)

2. In addition to the above point, the activity and specificity of sgRNA are also quite important to choose the target site. Is this point considered in MHcut tool? As the authors may know very well, there are a number of tools developed for scoring the sgRNA activity and specificity. Is it possible to implement such a function in the MHcut tool?

We thank Reviewer #2 for their suggestion to further improve the MHcut tool. There are indeed a number of tools that aim to predict the activity and specificity of guide RNAs. However, we feel that this is still a developing area of research, and the accuracy of activity and specificity predictions continues to undergo fundamental improvements (Huston et al., 2019, PMID:31225747). For this reason, we intentionally decided to not implement such a function directly in the MHcut tool, in order to extend its longevity. Instead, we provide MHcut users with an easy-to-use batch export function of the identified guide RNAs targeting the variant of interest. Users can then employ their state-of-the-art tool of choice to predict the most active and specific guide RNAs for their experiment. This workflow is illustrated in Figure 5, as introduced above.

3. In the population analysis displayed in Figure 2, the target sites containing various microhomology lengths resulting in various deletion

lengths were examined. However, all of them contained less than three
distances between two microhomologies. The current manuscript
contains only one data for longer distance (Figure S2). It seems
insufficient, because there are a wide variety of microhomologies
distances (Figure 2a). Various population analysis should be added for the
target sites containing longer distances. If it is difficult to efficiently create
mutations with longer distance in a template-free manner, this point
should be clearly described as a limitation of the authors' approach.

We thank Reviewer #2 for allowing us to justify our selection of target sites more
clearly. Indeed, variants shown in Figure 2a have a wide variety of microhomology
distances. This is however not representative of the overall MHcut dataset. Of the 11
million μ H-flanked deletion mutations, 9 million or 81% have a distance of 0-2 bp
(Fig. 1f), which corresponds to our selected targets. The subset of variants shown in
Figure 2a tends to have longer μ H distances due to the selection of variants with an
available unique NGG PAM site.

Reviewer #2 correctly points out the fact that target sites containing longer
distances between the microhomologies are expected to be more difficult to create.
Data supporting this expectation has been published, for example by Allen et al., 2018
(Ref. #11) and Ata et al. 2018 (Ref. #10). We agree that it is important to clearly
describe this potential limitation for creating the target deletion in hiPSCs using the
template-free method suggested in our paper, and performed data analysis and
experiments as outlined below.

First, we analyzed the inDelphi predictions for variants with various
microhomology distances (Supplementary Fig. 7a). Second, we conducted
experiments for 11 additional variants to recreate deletions with longer μ H distances
in hiPSCs (Supplementary Fig. 7b,c,d). While we could not define a clear maximum
distance with such a small number of targets, the high variability of guide RNA
efficiency in the experiments highlighted another limitation of our approach, the small
number of unique gRNAs available per variant.

We have added a section to the Results part describing the limitations of
precisely creating target variants with the MHcut method as follows (page 11, lines 6-
31):

**"Assessing the scope of targetable variants**

The length of heterology is known to negatively affect the efficiency of
MMEJ repair, with suggested limits ranging from 5 bp¹⁰ to around 15 bp¹¹. While
over 80% of the variants in the MHcut dataset are either directly abutted or have a
very short distance of 0-2 bp (Fig. 1f), the strict need for a unique Cas9 gRNA biases
the set of targetable variants towards larger distances, with only 45% of variants
having a μ H distance of 0-2 bp. We expect this bias to be mitigated by using
engineered Cas9 or orthologues with different PAM requirements.

To assess the impact of increasing μ H-distance on the efficiency of template-
free creation of the MHcut variants, we first analyzed the MHcut dataset with the
inDelphi editing prediction tool¹⁴. The mean prevalence predicted by inDelphi
(average score from available mESC, HCT116, HEK293, K562, U2OS cell data) for
all MHcut variants with available Cas9 gRNA indicates that, depending on μ H length,
μ H distances between 4 and 10 bp are expected to cause the MMEJ ratio of the target
deletion to drop below 20% on average (Supplementary Fig. 7a).

To validate this data in hiPSCs, we chose a set of candidate variants with μ H
distances between 3 and 10 bp (Supplementary Fig. 7b) from our short-listed
pathogenic variants (Fig. 2a), and transfected RNP complexes as described above (Fig.
2c). While we could not identify a clear limit for μ H-distance impacting the target
MMEJ ratio, we observed the highest MMEJ ratio (91%) for a non-annotated variant
in TSC2 with a μ H distance of 7 bp (Supplementary Fig. 7d). As observed for GLA
(Supplementary Fig. 2a), these data suggest that larger μ H distances are more likely
to contain nested μ Hs, interfering with target deletion creation. Importantly, the
results also show that low gRNA efficiencies are another major obstacle in creating
the target deletion (Supplementary Fig. 7c,d), supporting the need for reliable gRNA
prediction software or engineered Cas9 variants to fully explore the MHcut dataset.”

and have included points in the Discussion part as well (page 12, lines 19-25).

“The limited number of gRNAs per MHcut variant and the prevalence of the target
editing outcome, especially for variants with longer μ H distances, are a constraint on
the scope of targetable variants. We found gRNA efficiencies to be highly variable
and therefore recommend prioritizing the gRNAs identified by the MHcut tool not
only with tools for editing outcome prediction, but also with state-of-the-art gRNA
“on-“ and “off-target” efficiency evaluation software (Fig. 5) to maximize the
efficiency of target variant creation.”

Supplementary Figure 7. Assessing the influence of increasing μH distance on target MMEJ repair outcome efficiency.

(a) inDelphi¹⁴ predicted prevalence (average score from the 5 cell types tested) for all MHcut variants with available Cas9 guide RNA by μH-distance with μH length indicated by fill color. Dotted line represents 20% predicted prevalence.

(b) Selected target variant list. μH (green), DSB location (pink bolt), SpCas9 PAM (underline), Ye (guide RNA targeting CCR5 published by Ye et al., 2014), He (guide RNA targeting CCR5 presented by Jiankui He at the Second International Summit on Human Genome Editing in 2018).

**(c) Overall ratio of indel mutations found in the transfected hiPSC cell populations.**

**For COL1A, FLNA and TSC2 (10) nested μ Hs of 2 bp size reduce the target**
**MMEJ ratios. TTN has a 4 bp μ H outside of the target μ Hs with a higher GC**
**ratio that reduces the target MMEJ ratio. For Means \pm s.d. for n = 3 biological**
**replicates, $p < 0.05$. Dotted line represents 5% overall ratio of indel mutations.**

**(d) Ratio of the target MMEJ outcome among total indels. Targets in red font have**
**an overall indel ratio above 5%. Means \pm s.d. for n = 3 biological replicates, $p <$**
**0.05; mhDist (μ H distance); inDelphi (predicted prevalence of target MMEJ**
**outcome from inDelphi algorithm).**

4. The actual usefulness and importance of some selectable filters are
unclear. For example, two-cut option was provided, but there is no actual
example in the manuscript. Regarding the GC-content, there is a
suggestive description (line 225), but it is still unclear without showing
other examples.

We thank Reviewer #2 for raising the lack of clarity concerning the reason for why
we have provided the different filter options. To address this point, we have created
**Supplementary Table 1** explaining the meaning behind all MHcut output data we
created in addition to the data available from the dbSNP and ClinVar databases that
can be used as filter options. For example, the importance of the GC content has been
published by Shen et al., 2018 (Ref. #14), as part of the inDelphi scoring system, and
by Allen et al., 2018 (Ref. #11). The two-cut option has for example been published
by Guo et al., 2018 (PMID:30340517) and also by our group in the context of the
MMEJ-based MhAX cassette excision method (Kim S.I. et al., 2018, Ref. #13). The
strategy could be used to create variants with large distances between μ Hs. To verify
the impact of each μ H-feature on the repair outcome in hiPSCs, a large dataset is
necessary, which goes beyond the scope of this paper. Nevertheless, based on
previous publications, we believe that these features might be of interest to other
researchers utilizing the dataset. Thank you for allowing us to make this point clear to
future users of MHcut.

**Supplementary Table 1. MHcut outputs provided in addition to data from**
**ClinVar and dbSNP databases**

1

Category	Name	Explanation
Variant file		
IDs	id	Variant ID number assigned by MHeut
Microhomology (μH) features	mh_l	Length of full μH including mismatches (bp)
	mh_l1	Length of first stretch of perfect μH (bp)
	hom	Homology ratio of full μH
	mh_dist	Distance between the μHs based on mh_l (bp)
	mh_l1dist	Distance between the μHs based on mh_l1 (bp)
	mh_seq_1	Sequence of μH (left side)
	mh_seq_2	Sequence of μH (right side)
	nbmm	Number of mismatches in the μH
	mh_max_cons	Length of longest consecutive stretch of μH within full μH (bp)
	gc	GC content of μH (high GC content is associated with strong μHs ^{11,14})
	mh_score	μH score calculated by MHeut to decide which flank of the deletion is the stronger μH (number of mh_l + mh_l1)
	flank	Flank configuration chosen by MHeut (1: outer-inner, 2: inner-outer)
Guide RNA features	pam_mot	Number of NGG PAMs in a valid location between the μHs
	pam_uniq	Number of PAMs with a unique protospacer sequence in the genome
	guides_no_nmh	Number of guide RNAs with no nested μH
	guides_min_nmh	Number of nested μHs for guide RNA with the least nested μHs
	max_2cut_dist	Maximum distance between Cas9 cut sites available (bp) (it is possible to use two guide RNAs close to each μH to create large deletions ¹³)
Predicted Prevalence of Target Deletion	max_indelphi_freq_mean	Maximum prevalence predicted by inDelphi ¹⁴ for target deletion (mean of 5 cell types for best guide RNA)
	max_indelphi_freq_mesc	Max. prevalence predicted by inDelphi ¹⁴ for target deletion in mESCs
	max_indelphi_freq_u2os	Max. prevalence predicted by inDelphi ¹⁴ for target deletion in U2OS
	max_indelphi_freq_hek293	Max. prevalence predicted by inDelphi ¹⁴ for target deletion in HEK293
	max_indelphi_freq_het116	Max. prevalence predicted by inDelphi ¹⁴ for target deletion in HCT116
	max_indelphi_freq_k562	Max. prevalence predicted by inDelphi ¹⁴ for target deletion in K562
Visualization	cartoon	Cartoon showing the variant region with annotated μHs and cut sites
Corresponding Guides file		
IDs	id	Guide RNA ID number assigned by MHeut
	variant_id	Variant ID number assigned by MHeut
Guide RNA features	protospacer	Sequence of protospacer for Cas9 guide RNA
	mm0	Number of exact matches in the genome for the guide RNA
	m1_dist_1	Distance between cut site and perfect microhomology (left side) (bp)
	m1_dist_2	Distance between cut site and perfect microhomology (right side) (bp)
	mhdist_1	Distance between cut site and full microhomology (left side) (bp)
	mhdist_2	Distance between cut site and full microhomology (right side) (bp)
Predicted Prevalence of Target Deletion	indelphi_freq_mean	Prevalence predicted by inDelphi ¹⁴ for target deletion (mean of 5 cell types)
	indelphi_freq_mesc	Prevalence predicted by inDelphi ¹⁴ for target deletion in mESCs
	indelphi_freq_u2os	Prevalence predicted by inDelphi ¹⁴ for target deletion in U2OS cells
	indelphi_freq_hek293	Prevalence predicted by inDelphi ¹⁴ for target deletion in HEK293 cells
	indelphi_freq_het116	Prevalence predicted by inDelphi ¹⁴ for target deletion in HCT116 cells
	indelphi_freq_k562	Prevalence predicted by inDelphi ¹⁴ for target deletion in K562 cells
Nested μHs	nb_nmh	Number of nested μHs for this guide RNA
	largest_nmh	μH length for the longest nested μH (bp)
	nmh_size	μH length of the "best" nested μH (bp)
	nmh_var_l	Deletion length of the "best" nested μH (bp)
	nmh_seq	Sequence of the "best" nested μH
	nmh_gc	GC content of the "best" nested μH
	nmh_score	Microhomology-Predictor ¹⁵ score for the highest scoring "best" nested μH

2

3

5. The statement "data not shown" is prohibited in Nature Communications
(<https://www.nature.com/ncomms/submit/how-to-submit>). The
authors are requested to provide all the data.

Thank you for noting this oversight. We have prepared and submitted the SNP array
data in **Supplementary Figures 5c and 6c**.

**Supplementary Figure 5. Characterization of hiPSC DYSF-5bpDel clones.**

**(c) Karyogram showing all chromosomes of undifferentiated DYSF-5bpDel clones**
**and parental 1383D6 hiPSCs. B Allele Frequency (top) and Log R Ratio**
**(bottom) of SNP array (v1.2) analysis detect no large CNVs.**

 **Supplementary Figure 6. Characterization of hiPSC ALAS2-4bpDel and FECH-**
 **5bpDel disease model clones.**

**(c) Karyogram showing all chromosomes of undifferentiated ALAS2-4bpDel and**
 **FECH-5bpDel clones and parental 1383D6 hiPSCs. B Allele Frequency (top)**
 **and Log R Ratio (bottom) of SNP array (v1.2 and v1.3) analysis detect no large**
 **CNVs.**

 **Reviewer #3**

In this work, the authors describe a comprehensive computational analysis of
 microhomologies in the human genome, where they are widespread (highlights: in
 exons in most protein-coding genes, and surrounding the majority of annotated

deletion mutations). They describe a computational tool, MHCut, which is suggested
as a method for identifying candidates for precise repair to a target deletion allele via
microhomology-mediated end-joining (MMEJ). The authors demonstrate genotypic
validation of predicted precision editing in 7 disease-relevant deletion targets, and
characterize the efficiency of MMEJ in human induced pluripotent stem cells (hiPSC)
compared to HEK293T cells. The authors also perform cursory functional assays on
hiPSC cell lines generated from 3 disease-relevant precise deletion edits.

While some of their analysis is interesting, I think publishing this paper would
mislead the community, as MHCut is a strictly inferior tool to several recently
published algorithms aimed at predicting Cas9 outcomes (inDelphi, FORECasT from
Shen et al., 2018 and Allen et al., 2018). The above-referenced algorithms have been
trained and tested on extensive collections of Cas9 outcome profiles, while MHCut
seemingly does not use training data and uses a modest set of testing data.

Based on the work described, we believe that the authors overstate in the
abstract, introduction, and discussion the general extent to which MHCut is able to
identify candidates, and the extent to which candidates exist in the human genome, for
precise template-free editing. A representative text is "Here we introduce a tool called
MHCut that identified 11 million naturally occurring deletion mutations flanked by
μ Hs across the human genome, covering 88% of protein-coding genes. These
mutations are candidates for precise creation in a template-free manner by MMEJ
repair." in the abstract — the implication is that a substantial fraction of these 11
million deletion mutations identified by MHCut would be precisely created
experimentally, but the manuscript fails to support this claim.

If the authors believe that MHCut has advantages over inDelphi/FORECast
(which have user-friendly interfaces and can be easily utilized to give what are likely
to be more accurate estimates of genomic sites capable of precise Cas9-induced repair
to a mutation allele), then they should more clearly state and test such specific claims.
Otherwise, it will only confuse the research community to have a published tool that
has not been compared to the state-of-the-art, uses more vague terminology (the
authors never define precise repair, although the above-referenced papers give clear,
quantitative definitions) and is likely to be strictly inferior.

While I do not support revision of this manuscript for Nature Communications,
below I list some technical suggestions on other aspects of the work:

We thank Reviewer #3 for their constructive criticism of our manuscript, and clearly
highlighting the possibility for MHcut to be confused for a prediction algorithm. We
consider MHcut to be a classification and identification tool, distinguishing it from
tools such as inDelphi, which aim to predict editing outcomes. Because the goals of
these tools are different we do not believe they can be directly compared. They are,
however, complimentary, and we hope that incorporating inDelphi data into MHcut
and our practical application scheme (Fig. 5) will alleviate Reviewer #3's concerns
and the potential for reader confusion.

- 1. One claim of particular note is that the majority of pathogenic deletions in
humans possess microhomology. The authors should use simulated
random deletion data to derive background rates of random deletion
outcomes of particular lengths that would possess microhomology. This
would allow them to calculate the statistical significance of their claim
that microhomology deletions are enriched among human pathogenic
deletions.

We thank Reviewer #3 for allowing us to clarify the statistical significance of our
finding that the majority of known human deletion variants are flanked by
microhomologies. We have added an explanation to the results part (page 4-5, lines
33-4) for clarification, as follows: “Strikingly, of the 19.3 million deletion mutations
with a minimum size of 3 bp, 57% (11.1 million) are flanked by perfect μ Hs of at
least 3 bp size, far exceeding the probability expected by random base distribution
($0.25^3 = 1.56\%$). Surprisingly, for deletions of at least 1 or 2 bp size with at least 1 or
2 bp flanking μ Hs, homologous bases are detected in 75% (expected $0.25^1 = 25\%$)
and 67% (expected $0.25^2 = 6.25\%$) of variants, respectively (Supplementary Fig. 1a),
implicating microhomology as a common enriched characteristic of human annotated
deletion variants.”

- 2. An analysis of what fraction of pathogenic deletions could be created at
specific levels of precision (e.g. >50% of all genotypic outcomes) from
inDelphi/FORECasT would be a helpful addition, as it would likely be
much more accurate than MHCut and thus helpful to the community. This

might also clarify the failure of precise deletion at the KDM6A locus,
where the well-documented possibility of 1-bp insertions (Shen et al.,
2018, Allen et al., 2018, Kalhor et al., 2018, Lemos et al., 2018, Taheri-
Ghahfarokhi et al., 2018) confounds the precision of their targeted repair
(the authors do not describe MHCut as considering the possibility of 1-bp
insertions).

As Reviewer #3 suggests, our MHCut tool could be complemented by the addition of
the predicted repair outcome of the target deletion by published editing prediction
tools. MHCut only identifies existing deletion variants in the human genome that are
flanked by microhomologies and does not predict the prevalence of this outcome
among all possible editing outcomes. For this purpose, we have added the inDelphi
scores for all variants with available Cas9 guide RNAs to the MHCut dataset, enabling
users to filter for variants with a level of prevalence of their choosing. While the
predictions by inDelphi in the 5 available cell lines do not match our experimental
results in hiPSCs exactly, variants with a high inDelphi prediction are likely to have a
high prevalence. Nevertheless, we believe it is necessary to train inDelphi on a large
experimental dataset of repair outcomes in hiPSCs, in order to reliably use the scores
to predict editing precision.

As Reviewer #3 points out, indeed, inDelphi predicts the target deletion in the
KDM6A locus to only have a prevalence of 23% in the edited cell population (mean
across the 5 cell lines available in inDelphi). This is similar to our result of 29% in
hiPSCs. For the purposes of deriving gene-edited cell lines, 29% prevalence is
sufficient to recover a clone with the desired deletion with high confidence.

Out of interest, we performed the suggested analysis of the fraction of
pathogenic deletion mutations with available unique Cas9 guide RNAs that inDelphi
predicts (mean across the 5 cell lines) could be re-created at a specific level of
precision, in this case over 25%. Of the known pathogenic μ H-flanked variants that
can be targeted with a unique Cas9 gRNA (730), 271 or 37% have an inDelphi score
of >25%. Astonishingly, there are over 450,000 variants with a unique PAM and an
inDelphi score of over 25%, 14,649 of which are located in an exon.

We have added a short description of using the inDelphi score for defining the
limitations of precisely re-creating target variants with the MHCut method in the

Results part under the sub-heading of "Assessing the scope of targetable variants"
(page 11) and in Supplementary Figure 7.

3. In the three experiments where the authors installed biologically relevant
deletion mutations selected by their tool, the authors claimed that SNP
arrays enabled them to confirm the absence of additional CNVs or large
deletions in edited cells compared to the parental cell line, but the data is
not shown. The authors' claims would be better supported if the data
were shown. However, I have further concerns that a SNP array approach
is insufficient to fully confirm the authors' claim. A SNP array queries only
a tiny fraction of the genome, severely limiting the position and resolution
of detectable deletion events. Furthermore, in a diploid setting, only
deletions in heterozygous alleles would yield different SNP array results
compared to the parental cell line. To better support the works' claim that
useful disease models can be created by the authors' strategy, I would
suggest additional assays for measuring large deletions and additional
CNVs. In particular, large deletions at the expected cutsite could be
detected by targeted long-read sequencing (PacBio, Nanopore) or long-
range PCR (Kosicki et al., 2018).

We thank Reviewers #3 and #2 for noting this oversight. We have prepared and
submitted the SNP array data in Supplementary Figures 5c and 6c.

As Reviewer #3 astutely points out, we also acknowledge the limitation of the
SNP array of only being able to detect chromosomal abnormalities of more than 2 kb
size (median probe spacing) and have highlighted this in the Results part as follows:
"... and to have no large chromosomal abnormalities in comparison to parental
1383D6 hiPSCs by SNP array..." (page 8, lines 13-14), (page 9, lines 17-18) and
(page 10, lines 17-18). We have also added a section on the SNP array to Materials
and Methods as follows (Page 19, lines 10-19):

**SNP array**
Genomic DNA from iPSC clones modified by gene editing were genotyped using an
Infinium OmniExpress-24 v1.2 and v1.3 (Illumina) SNP array according to the
manufacturer's recommendations. Median probe spacing is around 2 kb. Array

scanning was performed on an iScan Bead Array Scanner (Illumina). Scanned data
were processed using Illumina GenomeStudio (2011.1 for v1.2 and 2.0.4 for v1.3)
with human genome Build 37 and FinalReports were exported according to the
manufacturer's protocol. CNV call was done using a combination of PennCNV53
(1.0.3), GWASTools54 (v1.2.R) and MAD55 (1.0.1). Karyograms were prepared
using GenomeJack (Mitsubishi Space Software).”

4. The authors state that only a small fraction of deletion mutations have
known phenotypes, yet all three mutations they model have known
phenotypes. It would have been more interesting to investigate mutations
without known functional effects. For example, the experiments in Fig. 3
are highly expected. Showing that a bi-allelic frameshift prevents protein
expression is not a novel or interesting finding. To support their claim
that precise, template-free deletions will enable novel approaches to
disease modeling, the paper should include novel biological analysis and
conclusions about at least one mutant allele created using this method.

We thank Reviewer #3 for drawing attention to the huge potential of investigating the
effect of variants of unknown significance (VUS). The current paper focusses on the
biological phenomenon of human deletion mutations being flanked by μ H and their
potential application for template-free genome editing. Therefore, we chose clear
examples for demonstration, and to highlight the fact that frameshift mutations do not
always result in a loss of protein expression or function, such as the case for ALAS2.
We agree that the MHcut dataset presents a unique opportunity for re-creating VUS in
the future. However, rather than fishing for one novel allele, this approach is best
addressed by building a phenotyping pipeline to screen the effects of a large number
of VUS in parallel; something we envision as a valuable yet independent study.

5. Lastly, the authors note in the discussion that "Furthermore, the MHcut
data and MMEJ could be applied to create protective alleles against
communicable disease, including the well-characterized 32 bp deletion in
CCR5 which confers resistance to HIV infection." This claim is poorly
supported by the work described and is inconsistent with the literature

on MMEJ. Among the 7 deletion targets described in the work, five are 4-5
nt deletions, and the longest is a 15-nt deletion. Though particular
outcomes in MMEJ are favored if they have strong microhomology, a
stronger factor is an exponentially decreasing frequency of repair of
longer deletions. As a result, a 32-bp deletion is expected to occur with
much lower precision, and the authors fail to provide evidence against
this expectation.

As Reviewer #3 astutely observed, the 32 bp deletion in CCR5 is predicted by
inDelphi to occur at very low frequency of just about 0.04% (mean across all cell
types) for the two available Cas9 guide RNAs that can introduce a DSB between the
μ Hs. Indeed, when we attempted to recreated the variant, the main repair outcome
was a +1 bp insertion. We could not detect the 32 bp deletion at significant levels
(Supplementary Fig. 7d), which matches with the inDelphi prediction. The large
distance between the μ Hs of 29 bp is probably the main factor in favoring the NHEJ
repair outcome over the MMEJ based deletion. This points to a clear limitation of our
approach that we have now clarified in the new Supplementary Figure 7 and in the
new Results part “Assessing the scope of targetable variants” (page 11), as described
above.

Accordingly, we deleted the following sentence from the Discussion section of
the manuscript: “Furthermore, the MHcut data and MMEJ could be applied to create
protective alleles against communicable disease, including the well-characterized 32
23 bp deletion in CCR5 which confers resistance to HIV infection⁴³”.

25 26 **ADDITIONAL MANUSCRIPT MODIFICATIONS**

1. The author affiliation for David Loughheed was changed to the Canadian Center
for Computational Genomics.
- 2. The abstract was modified (page 1, lines 5-10) to “Recently, the sequence context
surrounding nuclease-induced double strand breaks (DSBs) has been shown to
predict repair outcomes, for which μ H plays an important role. Building on this
observation, we surveyed naturally occurring human deletion variants and

- identified that 11 million or 57% are flanked by μ Hs, covering 88% of protein-
coding genes.”
- 3. Minor changes were made to the main text, including:
- a. Page 3, lines 16 and 28, page 9, line 14, page 10, line 15, “predictable”,
“predicted” changed to “precise”
- b. Page 5, line 10, “target space” changed to “variant number”
- c. Page 6, line 21, “and associated gRNAs” added
- 8 d. Page 12, lines 14-16, “The MHcut dataset can also contribute a relevant list of
9 endogenous target loci for creating training datasets for prediction algorithms
in the future.” added to Discussion part
- e. Page 14, lines 28-31, “In addition, MHcut calculates the predicted prevalence
of each variant among repair outcomes using the inDelphi algorithm¹⁴ for
each cut position and for each of the 5 cell types provided by inDelphi. The
predicted prevalences are recorded for each guide and summarized for the best
guide at the variant level.” added to Materials and Methods section
- f. Page 15, lines 3 and 8, dbSNP database updated to October 28, 2018 release.
- 17 g. Page 15, line 20, “and Excel” added
- 18 h. Page 23, lines 6-8, “and the data was deposited at
19 <https://doi.org/10.6084/m9.figshare.9118364>. The data that support the
20 findings of this study are also available from the corresponding author upon
request.” added to the Data Availability statement
- i. Page 23, lines 19-21, “The authors would like to thank Hiromi Dohi, Fumiyo
Kitaoka, Masaki Nomura, Tomoko Takahashi, Masafumi Umekage, and
Naoko Takasu for performing the SNP array analysis.” added to the
Acknowledgements section
- j. Page 23, lines 23-24, AMED grant numbers added for M.K.S. and H.S. to the
Acknowledgements section
- 4. Minor modifications to the Figures and Legends are listed:
- a. Updated design for Graphical Abstract
- b. In Figure 2 “Selected pathogenic” added to the title
- c. In Figure 2, panel a, reversed order of filters used to narrow down the dataset;
in legend added “Filtered MHcut tool output of potential target...” and
changed “Data” to “Graph”

- 1 d. In Figure 4, title changed to “Erythropoietic protoporphyria disease models
created by MMEJ display **ALAS2 gain-of-function and FECH loss-of-**
**function.**” to match Figure 3
- e. In Figure 4, panel e, “CD235a-**FITC**” and “Cd71-**APC**” added in legend;
panels g,h changed FACS plot axis label from “PPIX” to “PPIX-**Qdot® 605**”
in accordance with Journal’s guidelines.
- f. In Supplementary Figure 3 we updated the MHcut browser screenshots.
- 8 g. In Supplementary Figure 5 we deleted “disease model” from the title.
- 9 h. In Supplementary Tables 4 and 5 added crRNAs and primers used for revision
10 experiments
- 11 i. Adjusted numbering for figures and tables

REVIEWERS' COMMENTS:

Reviewer #1 (Remarks to the Author):

The authors have provided satisfactory responses to all of my queries and have revised the manuscript appropriately.

Reviewer #2 (Remarks to the Author):

The authors have responded to all the comments and revised the manuscript adequately. I have no further comments.

Reviewer #3 (Remarks to the Author):

The rebuttal submitted by Grajcarek et al does not substantively address any of the major critiques of this work. To reiterate these critiques:

1. MHCut is a strictly inferior tool to several recently published algorithms aimed at predicting Cas9 outcomes (inDelphi, FORECasT from Shen et al., 2018 and Allen et al., 2018). The above-referenced algorithms have been trained and tested on extensive collections of Cas9 outcome profiles, while MHCut seemingly does not use training data and uses a modest set of testing data. Based on the work described, we believe that the authors overstate in the abstract, introduction, and discussion the general extent to which MHCut is able to identify candidates, and the extent to which candidates exist in the human genome, for precise template-free editing. It is unclear what value is added by the authors' curation of a set of mutations with microhomologies when a collection of all ClinVar/HGMD mutations can be (and were in the Shen et al paper) directly fed into the user-friendly interfaces of published predictive algorithms of Cas9 outcomes with more trustworthy results. The authors do not address what deficiency in this pipeline their work addresses and thus what value it adds. If the only value added in this work is that the authors documented the prevalence of microhomologies in human mutations, this should be stated, and this does not seem impactful enough to be published in Nature Communications.
 2. To support their claim that precise, template-free deletions will enable novel approaches to disease modeling, the paper should include novel biological analysis and conclusions about at least one mutant allele created using this method. The authors sidestep this critique in their resubmission. Given that the computational component of their work is duplicative and inferior to previously published work, this uncovering of novel biology would seem a minimum to consider publication at Nature Communications.
- Altogether, the resubmission does not address either of the key weaknesses of the work, so I do not recommend publication.

NCOMMS-19-03896

Genome-wide microhomologies enable precise template-free editing of biologically relevant deletion mutations

Response to referees' comments

In light of Reviewer #3's continuing concerns, we ask that the differences between MHCut and other bioinformatics tools, such as inDelphi, are discussed - particularly in terms of design and applicability – as well as a discussion regarding how MHCut complements existing approaches.

→ We have added further explanations on the differences and complementarity between MHCut and other bioinformatics tools, such as inDelphi, to the Discussion section. Below is our concrete response to the Reviewers comments.

Reviewer #1 (Remarks to the Author):

The authors have provided satisfactory responses to all of my queries and have revised the manuscript appropriately.

Reviewer #2 (Remarks to the Author):

The authors have responded to all the comments and revised the manuscript adequately. I have no further comments.

Reviewer #3 (Remarks to the Author):

The rebuttal submitted by Grajcarek et al does not substantively address any of the major critiques of this work. To reiterate these critiques:

1. MHCut is a strictly inferior tool to several recently published algorithms aimed at predicting Cas9 outcomes (inDelphi, FORECasT from Shen et al., 2018 and Allen et al., 2018). The above-referenced algorithms have been trained and tested on extensive collections of Cas9 outcome profiles, while MHCut seemingly does not use training data and uses a modest set of testing data. Based on the work described, we believe that the authors overstate in the abstract, introduction, and discussion the general extent to which MHCut is able to identify candidates, and the extent to which candidates exist in the human genome, for precise template-free editing. It is unclear what value is added by the authors curation of a set of mutations with microhomologies when a collection of all ClinVar/HGMD mutations can be (and were in the Shen et al paper) directly fed into the user-friendly interfaces of published predictive algorithms of Cas9 outcomes with more trustworthy results. The authors do not address what deficiency in this pipeline their work addresses and thus what value it adds. If the only value added in this work is that the authors documented the prevalence of microhomologies in human mutations, this should be stated, and this does not seem impactful enough to be published in Nature Communications.

→ MHCut is not an algorithm for predicting DNA repair outcomes and as such does not compete with, but complements the existing prediction algorithms (revised Figure 5). Shen et al (2018) did use inDelphi to look at ClinVar/HGMD mutations in order to evaluate the possibility of re-establishing reading frame or remove duplications, irrespective of the repair pathway (NHEJ or MMEJ). Our approach differs in that MHCut starts by analyzing the sequence of a variant allele, identifies existing μ H at the deletion ends, and then identifies gRNAs targeting the reference genome that could potentially be used to create the deletion variant through MMEJ. None of the recent prediction tools are designed to investigate the extent of microhomologies in the genome and cannot replicate the MHCut dataset.

MHCut indeed identified the prevalence of microhomologies flanking human mutations and we have shown that this enables a strategy for editing endogenous loci based on MMEJ repair. Gaging from the positive responses we've received at various scientific conferences, we believe the discovery of this natural phenomenon to be of interest not only to the wider genome editing community, but also to those researching genome biology. The added value of MHCut for the research community is that researchers can search the library of microhomology-flanked biologically-relevant mutations existing in the human genome for variants of interest, before using the prediction tools like inDelphi or FORECasT to predict the most efficient candidates.

2. To support their claim that precise, template-free deletions will enable novel approaches to disease modeling, the paper should include novel biological analysis and conclusions about at least one mutant allele created using this method. The authors sidestep this critique in their resubmission. Given that the computational component of their work is duplicative and inferior to previously published work, this uncovering of novel biology would seem a minimum to consider publication at Nature Communications. Altogether, the resubmission does not address either of the key weaknesses of the work, so I do not recommend publication.

→ A novel biological analysis of mutant alleles would require establishing a high-throughput phenotyping assay (likely focused on a specific gene family) to find a proverbial "needle in the haystack". We are preparing for such an assay, however it could take many months or even years to establish correctly. For that reason, we want to make the library of variants available to all researchers as soon as possible, and therefore believe this request to be out-of-scope for the current paper. Similar to papers from Josef Penniger's Lab "Forward and Reverse Genetics through Derivation of Haploid Mouse Embryonic Stem Cells" (DOI: 10.1016/j.stem.2011.10.012) and

“Comparative glycoproteomics of stem cells identified new players in ricin toxicity” (DOI: 10.1038/nature24015), we think separating the library dataset from the specific phenotyping assay to be the most suitable strategy for publication. I would like to point out that in the inDelphi study, Shen et al. (2018) did not include any novel biological analyses, only proof-of-principle for their approach. Note that in recreating the ALAS2 variant, we generated the first iPSC model of X-linked protoporphyria, demonstrating that frameshift mutations can result in gain-of-function phenotypes, challenging the broadly used strategy of generating frameshifts for gene knockout.